# R-loop formation and conformational activation mechanisms of Cas9

Martin Pacesa[1], Luuk Loeff[1,2], Irma Querques[1,2], Lena M. Muckenfuss[1,2], Marta Sawicka[1] & Martin Jinek[1✉]

Cas9 is a CRISPR-associated endonuclease capable of RNA-guided, site-specific DNA cleavage[1–3]. The programmable activity of Cas9 has been widely utilized for genome editing applications[4–6], yet its precise mechanisms of target DNA binding and off-target discrimination remain incompletely understood. Here we report a series of cryo-electron microscopy structures of *Streptococcus pyogenes* Cas9 capturing the directional process of target DNA hybridization. In the early phase of R-loop formation, the Cas9 REC2 and REC3 domains form a positively charged cleft that accommodates the distal end of the target DNA duplex. Guide–target hybridization past the seed region induces rearrangements of the REC2 and REC3 domains and relocation of the HNH nuclease domain to assume a catalytically incompetent checkpoint conformation. Completion of the guide–target heteroduplex triggers conformational activation of the HNH nuclease domain, enabled by distortion of the guide–target heteroduplex, and complementary REC2 and REC3 domain rearrangements. Together, these results establish a structural framework for target DNA-dependent activation of Cas9 that sheds light on its conformational checkpoint mechanism and may facilitate the development of novel Cas9 variants and guide RNA designs with enhanced specificity and activity.

Cas9 enzymes rely on a dual guide RNA structure consisting of a CRISPR RNA (crRNA) guide and a *trans*-activating CRISPR RNA (tracrRNA) coactivator to cleave complementary DNA targets. *S. pyogenes* Cas9 (SpCas9) has found widespread use as a programmable DNA-targeting tool in genome editing and gene-targeting applications[4–6]. Target DNA binding by SpCas9 is dependent on the initial recognition of an NGG protospacer-adjacent motif (PAM) downstream of the target site[2,7–9], which triggers local DNA strand separation to initiate its directional hybridization with a 20-nt segment in the guide crRNA to form an R-loop structure[7,10,11]. Target strand (TS) binding is facilitated by structural pre-ordering of nucleotides 11–20 of the crRNA (counting from the 5′ end), termed the seed sequence, in an A form-like conformation[8,12]. Formation of a complete R-loop leads to the activation of the Cas9 HNH and RuvC nuclease domains to catalyse cleavage of the TS and non-target DNA strand (NTS), respectively[2,8,13]. Although highly specific, SpCas9 cleaves off-target sites with imperfect complementarity to the guide RNA, often resulting in considerable levels of off-target genome editing[14–18]. The off-target activity is dependent on the number, type and positioning of base mismatches within the guide–target heteroduplex[15,19–21]. PAM-proximal mismatches within the seed region are discriminated against by substantially increased dissociation rates[11,19,21,22], whereas PAM-distal mismatches are compatible with stable DNA binding[13,19,21,23,24]. Such off-targets are instead discriminated by a conformational checkpoint mechanism that monitors the integrity of the guide–target duplex to induce conformational activation of the nuclease domains[11,13,19,21–24]. Structural, biophysical and computational studies of SpCas9 have

shed light on the mechanism of guide RNA binding, PAM recognition and nuclease activation, revealing that the enzyme undergoes extensive conformational rearrangements throughout these steps. In particular, high-resolution structures of the fully bound target DNA complex of SpCas9[25–28] have revealed a target-DNA-dependent conformational rearrangement of the Cas9 REC lobe that is necessary for cleavage activation. However, the mechanisms that underpin R-loop formation and off-target discrimination during conformational activation have remained elusive.

## Cryo-EM analysis of R-loop formation

To investigate the mechanism of R-loop formation, we initially determined the minimal extent of target DNA complementarity necessary for stable binding using fluorescence-coupled size-exclusion chromatography, revealing that the presence of six complementary nucleotides in the PAM-proximal region of the target DNA heteroduplex is sufficient for stable association with the SpCas9–guide RNA complex. (Extended Data Fig. 1). Subsequently, catalytically inactive SpCas9 (dCas9) was reconstituted with a single-molecule guide RNA (sgRNA) and partially matched DNA targets containing 6, 8, 10, 12, 14 and 16 complementary nucleotides upstream of the PAM (Fig. 1a and Extended Data Fig. 2). We analysed the resulting complexes using cryo-electron microscopy (cryo-EM), yielding molecular reconstructions at resolutions of 3.0–4.1 Å (Extended Data Fig. 3 and Extended Data Tables 1 and 2). We additionally determined cryo-EM reconstructions of wild-type

[1]Department of Biochemistry, University of Zurich, Zurich, Switzerland. [2]These authors contributed equally: Luuk Loeff, Irma Querques, Lena M. Muckenfuss. ✉e-mail: jinek@bioc.uzh.ch

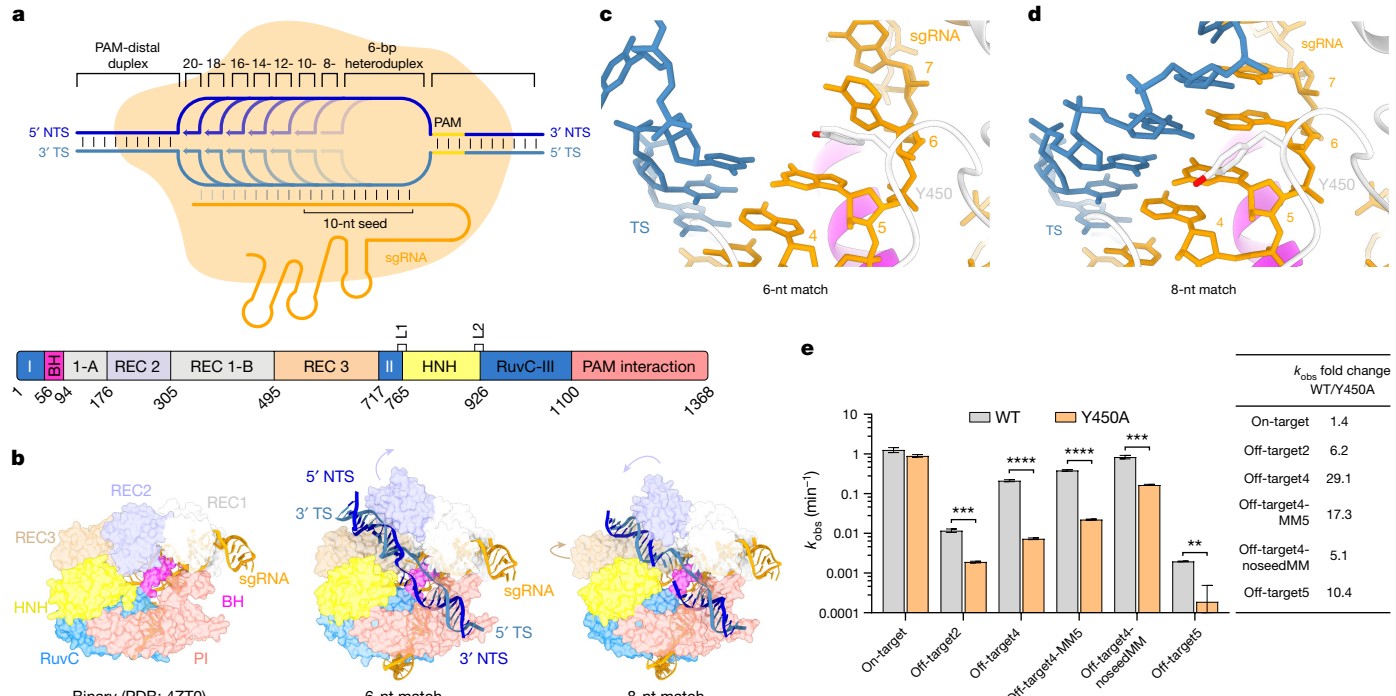

**Fig. 1 | Target DNA binding induces Cas9 REC lobe restructuring. a**, Top, schematic depicting DNA-bound complexes with increasing extent of complementarity to guide RNA. Bottom, domain composition of SpCas9. 1-A, REC1-A domain; I–III, RuvC domain motifs I–III; BH, bridge helix. **b**, Structural comparison of the SpCas9 binary (left), 6-nt match (middle) and 8-nt match (right) complexes. **c**, Zoomed-in view of the seed region of the guide RNA–target DNA heteroduplex in the 6-nt match complex. Tyr450 stacks between the fifth and sixth nucleotide, counting from the PAM-proximal end of the heteroduplex. **d**, Zoomed-in view of the seed region of the guide RNA–target DNA heteroduplex in the 8-nt match complex. **e**, Fitted cleavage rate ($k_{obs}$) of wild-type (WT) and Y450A mutant Cas9 against on-target and off-target substrates. Data represent mean fit ± s.e.m. of $n = 4$ independent replicates. Two-tailed $t$-test, ****$P < 0.0001$, ***$P = 0.0002$, **$P = 0.0011$. The $P$-value for the on-target dataset was not significant ($P = 0.1058$).

SpCas9 bound to 18-nt complementary DNA targets in the presence of 1 mM and 10 mM $Mg^{2+}$, representing the checkpoint and catalytically active states, respectively (Extended Data Fig. 3 and Extended Data Table 2). Three-dimensional variability analysis[29] was used to analyse conformational heterogeneity within each complex (Supplementary Videos 1–8). Most of the detected variability within each complex can be attributed to the PAM-distal duplex and the REC2, REC3 and HNH domains, suggestive of conformational equilibrium sampling. The resulting structural models are representative of the most abundant conformational state of each complex (Extended Data Fig. 4).

Structural superpositions of the partially matched complexes with the guide-RNA-bound binary SpCas9 complex[12] provide a framework for the visualization of the DNA-binding mechanism, revealing stepwise domain rearrangements coupled to R-loop formation (Extended Data Fig. 5a). All complexes exhibit almost identical conformations of the bridge helix, REC1, RuvC and PAM-interaction domains, as well as the PAM-proximal double stranded DNA (dsDNA) duplex and the sgRNA downstream (3′ terminal) of the seed region. Conformational differences are observed in the positioning of the REC2, REC3 and the HNH domain relative to the emerging R-loop, consistent with the 3D variability analysis.

## R-loop initiation by bipartite seed

The structure of the 6-nucleotide complementary target (6-nt match) complex reveals a 5-bp heteroduplex formed by the sgRNA seed sequence and TS DNA (Fig. 1b). Hybridization beyond the fifth seed sequence nucleotide is precluded by base stacking with the side chain of Tyr450, which was previously observed in the structure of the Cas9–sgRNA binary complex[12] (Fig. 1c). Comparisons with the binary complex structure indicate that TS hybridization is associated with the displacement of the REC2 domain out of the central binding channel

(Fig. 1b). The PAM-distal duplex part of the DNA substrate is bound in a positively charged cleft formed by the REC2 and REC3 domains (Fig. 1b and Extended Data Fig. 5b), stabilized by interactions of the REC2 residues Ser219, Thr249 and Lys263 with the NTS backbone (Extended Data Fig. 5c), and REC3 residues Arg586 and Thr657 with the TS backbone (Extended Data Fig. 5d). Similar REC lobe conformation and protein contacts with the PAM-distal end of the DNA have been observed in a 3-bp heteroduplex complex described in a recent study[30]. Consequently, the NTS is positioned parallel to the guide RNA–TS DNA heteroduplex within the central binding channel (Fig. 1b). The 5′-terminal part of the sgRNA appears to be conformationally flexible but residual cryo-EM density suggests its placement in a positively charged cleft located between the HNH and PAM-interaction domains (Extended Data Fig. 5e).

The structure of the 8-nucleotide complementary target (8-nt match) complex reveals that expansion of the R-loop heteroduplex, enabled by unstacking of Tyr450, forces further repositioning of the REC2 and REC3 domains to widen the binding channel as the PAM-distal duplex shifts deeper inside the channel (Figs. 1d and 2a–c and Extended Data Fig. 5f). R-loop propagation and PAM-distal duplex displacement results in the formation of new intermolecular contacts, with Cas9 contacting the PAM-distal duplex backbone through REC2 domain residues Ser217, Lys234 and Lys253, and REC3 residues Arg557 and Arg654 (Extended Data Fig. 5g,h).

Together, these observations suggest that the seed sequence of the Cas9 guide RNA is bipartite and that its hybridization with target DNA proceeds in two steps, consistent with the existence of a short-lived intermediate state observed in FRET studies[11,31]. To validate the observed interactions, we tested the cleavage activities of structure-based Cas9 mutant proteins in vitro (Extended Data Fig. 6a). Alanine substitution of Tyr450 resulted in substantial reductions of off-target substrate

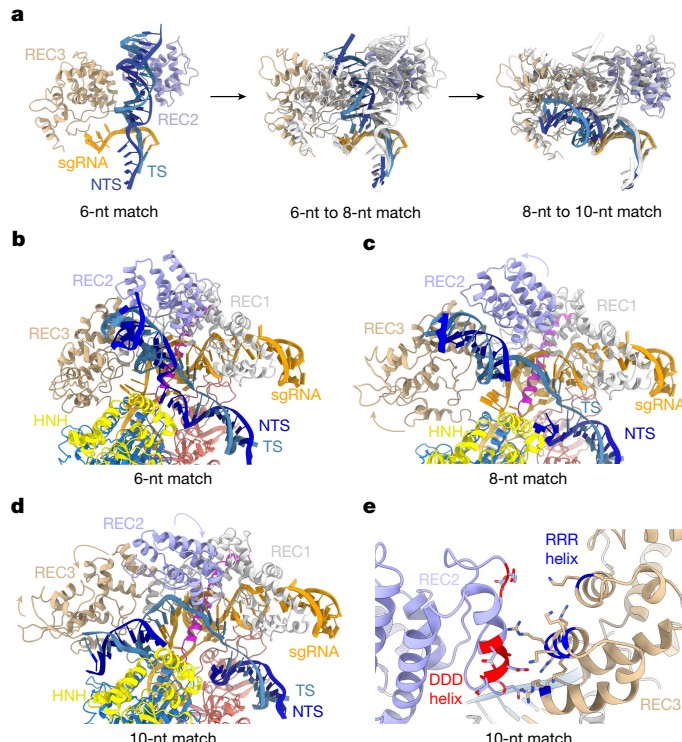

**Fig. 2 | R-loop propagation drives DNA repositioning within Cas9.**
**a**, Zoomed-in views of the conformational transitions in the PAM-distal DNA duplex and Cas9 REC2 and REC3 domains in the 6-, 8- and 10-nt match complex. **b**, Zoomed-in view of the R-loop in the 6-nt match complex. **c**, Zoomed-in view of the R-loop in the 8-nt match complex. **d**, Zoomed-in view of the R-loop in the 10-nt match complex. **e**, Zoomed-in view of the interaction between the REC2 domain DDD helix and the REC3 RRR helix.

cleavage rates, whereas on-target cleavage remained largely unperturbed (Fig. 1e and Extended Data Fig. 6b). As observed previously[32], the effect was more prominent for off-target substrates containing mismatches with the seed region of the guide RNA compared with off-targets containing only PAM-distal mismatches. Together, these results suggest that disruption of seed sequence interactions in the binary Cas9–sgRNA complex and early binding intermediates might exacerbate R-loop destabilization caused by off-target mismatches, resulting in an increased rate of off-target substrate dissociation and thus increased specificity. By contrast, a subset of mutations of DNA-interacting REC2 or REC3 residues resulted in increased off-target cleavage, as did the deletion of the REC2 domain (Extended Data Fig. 6b–e), consistent with single-molecule studies implicating the REC2 domain in Cas9 specificity[31]. Collectively, these results underscore the importance of specific Cas9–DNA contacts during early steps of R-loop formation for the specificity of Cas9.

## R-loop propagation and remodelling

Further guide RNA–TS hybridization to form a 10-bp heteroduplex causes a rearrangement of the REC2 and REC3 domains and repositioning of the PAM-distal DNA duplex into the positively charged central binding channel formed by the REC3, RuvC and HNH domains (Fig. 2a). Here, the PAM-distal dsDNA duplex forms a continuous base stack with the sgRNA–TS heteroduplex (Fig. 2d). The displaced NTS is positioned underneath the HNH domain and continues to run parallel to the extending guide RNA–TS DNA heteroduplex (Extended Data Fig. 7a). X-ray crystallographic analysis of the 10-nt match complex at a resolution of 2.8 Å (Extended Data Table 3) confirmed that the TS and NTS remain hybridized at the

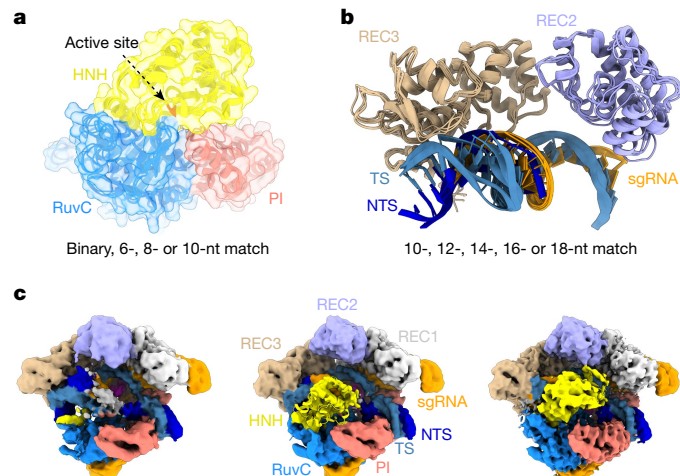

**Fig. 3 | Target pairing past the seed region undocks the HNH nuclease domain. a**, Position of the HNH catalytic site in the binary, 6-, 8- and 10-nt match complexes. **b**, Structural overlay of the REC2 and REC3 domains in the 10-, 12-, 14-, 16- and 18-nt match (checkpoint) complexes. **c**, Overview of the 12-nt match (left), 14-nt match (middle) and 16-nt match (right) complexes, shown in the same orientations. For each complex, the unsharpened cryo-EM map is overlaid with the respective atomic model. The 12-nt match complex map shows residual density for the displaced NTS (white). The 14-nt match map reveals residual density corresponding to the HNH domain. No density is visible for NTS. Cryo-EM maps are coloured according the schematic in Fig. 1a.

PAM-distal end of the DNA substrate (Extended Data Fig. 7b). The PAM-distal duplex is wedged between the REC3 and RuvC domains and the L1 HNH linker (Fig. 2d and Extended Data Fig. 7a,b). The relocation of the PAM-distal duplex causes the REC2 domain to shift closer to the binding channel and occlude the cleavage site in TS DNA (Fig. 2d). This shift also establishes new electrostatic interactions between a negatively charged helix in REC2 (Glu260, Asp261, Asp269, Asp272, Asp273, Asp274 and Asp276) and a positively charged helix in REC3 (Lys599, Arg629, Lys646, Lys649, Lys652, Arg653, Arg654 and Arg655), hereafter referred to as the DDD and RRR helices, respectively (Fig. 2e), which are highly conserved across Cas9 orthologues that contain a REC2 domain (Extended Data Fig. 7c). Cleavage of off-target substrates in vitro was reduced by alanine substitutions of the interacting residues in the REC2 DDD helix, whereas mutations in the REC3 RRR helix only reduced cleavage of the off-target substrate containing a mismatch in the seed region (Extended Data Fig. 6d,e). These results suggest that the REC2–REC3 interaction contributes to Cas9 restructuring during R-loop extension; however, the DDD and RRR helices might have additional structural roles during upstream and downstream steps in the DNA-binding mechanism, particularly as the REC3 RRR helix contacts the backbone of the PAM-distal DNA duplex during early stages of target binding (Extended Data Fig. 5d,h).

The HNH nuclease domain remains docked on the RuvC and PI domains in the 6-, 8- and 10-nt match complexes, with the active site buried at the interdomain interface (Fig. 3a). R-loop extension past the seed region to form a 12-bp heteroduplex does not result in major REC2/3 domain rearrangements, with the PAM-distal duplex remaining coaxially stacked onto the guide RNA–TS DNA heteroduplex throughout the 12-, 14-, 16- and 18-nt match complexes (Fig. 3b). By contrast, the HNH domain becomes disordered along with the surrounding RuvC 1011–1040 and PI 1245–1251 loops in the 12-nt match complex (Fig. 3c). Upon extension of the R-loop heteroduplex to 14 bp, the RuvC and PI loops responsible for HNH docking remain structurally disordered (Fig. 3c and Extended Data Fig. 8a) and residual density

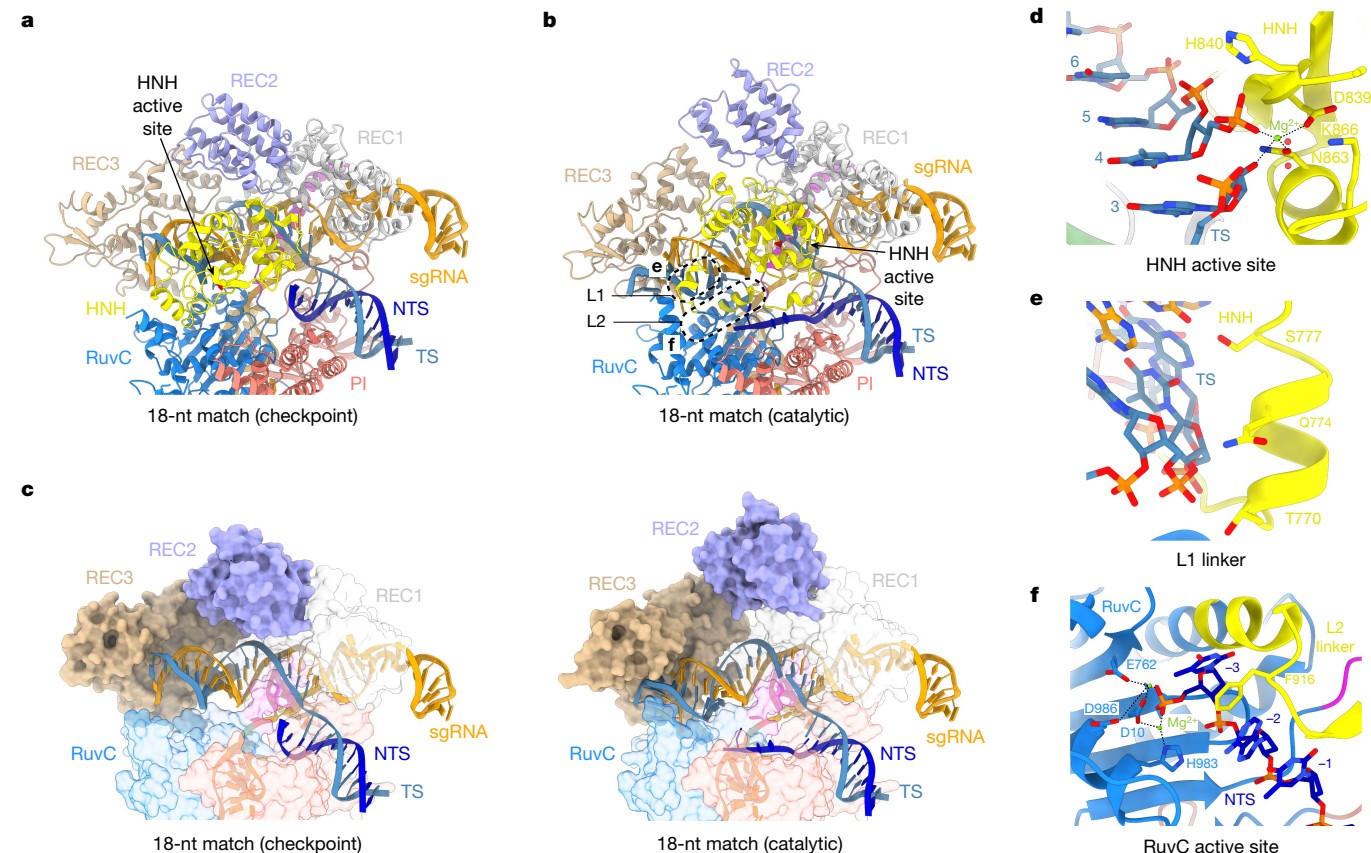

**Fig. 4 | HNH domain rotation and DNA bending enable catalytic activation.**
**a**, The structure of the 18-nt match complex in the pre-cleavage, checkpoint state. **b**, The structure of the 18-nt match complex in the catalytically active state. **c**, Conformations of the guide–target heteroduplexes and REC2 and REC3 domains in the 18-nt match checkpoint (left) and catalytic (right) complexes. The structures are shown in the same orientations as in **a**,**b**. The HNH domain has been omitted from the images for clarity. **d**, Zoomed-in view of the HNH nuclease active site in the 18-match catalytic complex containing bound cleaved TS. **e**, Zoomed-in view of the L1 linker contacting the minor groove of the guide RNA–target DNA heteroduplex. **f**, Zoomed-in view of the RuvC nuclease active site containing the 3′-terminal product of cleaved NTS.

is observed for the HNH domain as its L2 linker contacts the guide RNA–TS heteroduplex (Extended Data Fig. 8a). Further extension of the R-loop heteroduplex from 14 to 16 bp causes translocation of the HNH domain towards the guide RNA–TS DNA heteroduplex within the central binding channel (Fig. 3c). Facilitated by the formation of the PAM-distal part of the R-loop, a loop in the RuvC domain (residues 1030–1040) restructures into a helical conformation, establishing interactions with the L2 linker (Extended Data Fig. 8b). This repositions the L2 linker and shifts the HNH domain on top of the heteroduplex, sealing off the central binding channel (Fig. 3c and Extended Data Fig. 8c). The HNH domain remains in a catalytically incompetent orientation, with its active site located around 31 Å away from the scissile phosphate group in the TS.

## Conformational checkpoint and activation

Previous studies have shown that substrates containing 4-bp mismatches at the PAM-distal end of the target sequence (positions 16–20) are generally refractory to Cas9 cleavage, whereas substrates containing mismatches at positions 19 and 20 are efficiently cleaved[13,23,24,33]. The cryo-EM reconstruction of the 18-nt match complex in the presence of 1 mM Mg[2+] reveals that the most populated 3D class in the sample represents a pre-cleavage state with an intact TS and disordered NTS (Fig. 4a). Upon extension of the R-loop to 18 bp, the HNH domain continues to assume the catalytically incompetent orientation observed in the 16-nt match complex, whereas the conformation of the REC2 and

REC3 domains remains the same as in the 12-, 14- and 16-nt match complexes (Figs. 3b and 4a). The observed conformation is thus consistent with a catalytically inactive checkpoint state inferred from previous biophysical and structural studies[23,24,33].

The cryo-EM reconstruction obtained from a sample reconstituted in the presence of 10 mM Mg[2+] reveals a catalytically active conformation in which both the TS and the NTS are cleaved at the expected positions (Fig. 4b–f). In contrast to previously reported structures of catalytically active Cas9 enzymes[28,34,35], the PAM-proximal part of the cleaved NTS remains bound in the RuvC active site (Fig. 4b,f). In this state, the REC2 domain is shifted away from the TS cleavage site, enabling the HNH domain to undergo a rotation of about 140° to engage the TS scissile phosphate with its active site and catalyse its hydrolysis via a one-metal-ion mechanism (Fig. 4d), in agreement with previous structural data[28,34,35]. This rearrangement is facilitated by pronounced bending of the PAM-distal region of the guide RNA–TS DNA heteroduplex and a concomitant reorientation of the REC3 domain that preserves interactions with the heteroduplex (Fig 4c). HNH domain rotation is brought about by restructuring of the L1 and L2 linkers, which results in the widening of the NTS binding cleft and exposure of the RuvC active site (Fig. 4b,e,f). The L1 linker, which is structurally disordered in the 18-nt match checkpoint complex, forms an α-helix and interacts with the minor groove of the guide RNA–TS DNA heteroduplex via multiple hydrogen-bonding interactions (Fig. 4e). The L2 linker helix becomes extended, allowing Phe916 to intercalate between NTS nucleobases by π–π stacking, thereby stabilizing the NTS in the RuvC active site

(Fig. 4f). The NTS scissile phosphate is coordinated by two $Mg^{2+}$ ions, its position consistent with a His983-dependent catalytic mechanism proposed by molecular dynamics simulations[36]. A recent complementary study reported the structure of a 17-nt match catalytic complex that exhibits nearly identical HNH domain positioning and bent conformation of the guide RNA–TS DNA heteroduplex as observed in the 18-nt match catalytic complex[37], indicating that catalytic activation can occur once a 17-bp heteroduplex is formed. Together, these structural observations provide a rationale for the allosteric coupling of R-loop formation with HNH domain rearrangement and RuvC active site accessibility, in agreement with single-molecule studies showing that PAM-distal end positioning modulates HNH domain conformation[33].

## Conclusions

In sum, our structural analysis of SpCas9 along its DNA-binding pathway points to a mechanism whereby R-loop formation is allosterically and energetically coupled to domain rearrangements necessary for nuclease domain activation (Extended Data Fig. 9). The initial phase of R-loop formation is facilitated by TS hybridization to a bipartite seed sequence of the guide RNA and interactions of the PAM-distal DNA with the Cas9 REC2 and REC3 domains. The observation of a bipartite seed sequence in the Cas9 guide RNA and a two-step seed hybridization mechanism involving a conformational rearrangement brings parallels with other RNA-guided nucleic acid-targeting systems including the Cascade complex and Argonaute proteins, both of which feature discontinuous seed sequences in their guide RNAs[38–41]. We identify mutations that destabilize the binding intermediate states and thus increase off-target discrimination, which presents an opportunity for the development of novel high-fidelity SpCas9 variants. As most off-target sequences are only bound but not cleaved[19–21,42], these variants could prove useful for applications that rely on the fidelity of Cas9 target binding, such as transcriptional regulation or base editing[43]. Directional target DNA hybridization is associated with dynamic repositioning of the REC2, REC3 and HNH domains to initially assume a catalytically inactive, checkpoint conformation upon R-loop completion. As conformational activation of the nuclease domains is allosterically controlled by structural distortion of the PAM-distal end of the guide–target heteroduplex and the sensing of its integrity by Cas9, it is precluded by incomplete PAM-distal heteroduplex pairing (<17 bp). Bona fide off-target substrates are able to pass the conformational checkpoint because they maintain heteroduplex integrity despite the presence of PAM-distal mismatches, in agreement with our recent structural data[44]. Furthermore, guide RNA modifications that result in altered heteroduplex conformation have profound effects on Cas9 nuclease activity and specificity[45]. Together, our structural studies thus highlight the importance of maintaining guide–target complementarity and proper heteroduplex geometry, consistent with biophysical and computational studies showing that the conformation of the R-loop heteroduplex strongly affects off-target binding[11,46]. These findings thus have important implications for ongoing experimental and computational studies of CRISPR–Cas9 off-target activity, and will inform its further technological development.

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

## Methods

### Expression and purification of Cas9 proteins

Wild-type and mutant SpCas9 proteins were expressed in *Escherichia coli* Rosetta 2 (DE3) (Novagen) for 16 h at 18 °C as fusion proteins with an N-terminal His$_6$–MBP–TEV tag. Bacterial pellets were resuspended and lysed in 20 mM HEPES-KOH pH 7.5, 500 mM KCl, 5 mM imidazole, supplemented with protease inhibitors. Cell lysates were clarified using ultracentrifugation and loaded on a 15 ml Ni-NTA Superflow column (QIAGEN) and washed with 7 column volumes of 20 mM HEPES-KOH pH 7.5, 500 mM KCl, 5 mM imidazole. Tagged Cas9 was eluted with 10 column volumes of 20 mM HEPES-KOH pH 7.5, 250 mM KCl, 200 mM imidazole. Salt concentration was adjusted to 250 mM KCl and the protein was loaded on a 10 ml HiTrap Heparin HP column (GE Healthcare) equilibrated in 20 mM HEPES-KOH pH 7.5, 250 mM KCl, 1 mM DTT. The column was washed with 5 column volumes of 20 mM HEPES-KOH pH 7.5, 250 mM KCl, 1 mM DTT, and dCas9 was eluted with 15 column volumes of 20 mM HEPES-KOH pH 7.5, 1.5 M KCl, 1 mM DTT, in a 0–50% gradient (peak elution around 500 mM KCl). His$_6$–MBP tag was removed by TEV protease cleavage overnight at 4 °C with gentle shaking. The untagged protein was concentrated and further purified on a Superdex 200 16/600 gel filtration column (GE Healthcare) in 20 mM HEPES-KOH pH 7.5, 500 mM KCl, 1 mM DTT. Pure fractions were concentrated to 10 mg/ml, flash frozen in liquid nitrogen and stored at 80 °C.

### sgRNA in vitro transcription

The sgRNA was transcribed from a dsDNA template (Supplementary Table 1) in a 5 ml transcription reaction (30 mM Tris-HCl pH 8.1, 25 mM MgCl2, 2 mM spermidine, 0.01% Triton X-100, 5 mM CTP, 5 mM ATP, 5 mM GTP, 5 mM UTP, 10 mM DTT, 1 μM DNA transcription template, 0.5 units inorganic pyrophosphatase (Thermo Fisher), 250 μg T7 RNA polymerase). The transcription reaction was incubated at 37 °C for 5 h, after which the dsDNA template was degraded for 30 min with 15 units of RQ1 DNAse (Promega). The transcribed sgRNA was PAGE purified on an 8% denaturing polyacrylamide gel containing 7 M urea, ethanol precipitated and dissolved in DEPC-treated water.

### Gel filtration binding assay

The dCas9–guide RNA complex was assembled by incubating 371 pmol dCas9 with 400 pmol of the sgRNA in 20 mM HEPES-KOH pH 7.5, 200 mM KCl, 2 mM MgCl$_2$ for 10 min at room temperature. Then 250 pmol of Cy5-labelled dsDNA substrate was added and incubated another 15 min. The volume was adjusted up to 100 μl with reaction buffer and the mixture was centrifuged to remove possible precipitates. Individual reactions were transferred to a 96-well plate and analysed using a Superdex 200 Increase 5/150 GL gel filtration column (GE Healthcare) attached to an Agilent 1200 Series Gradient HPLC system. The 260 nm, 280 nm and Cy5 signals were exported and plotted as a function of the retention volume in GraphPad Prism 9.

### In vitro nuclease activity assays

Cleavage reactions were performed at 37 °C in reaction buffer, containing 20 mM HEPES pH 7.5, 250 mM KCl, 5 mM MgCl$_2$ and 1 mM DTT. First, Cas9 protein was pre-incubated with sgRNA in 1:1.25 ratio for 10 min at room temperature. The protein–RNA complex was rapidly mixed with the dsDNA substrates (containing 5′-ATTO-532 labelled TS) (Supplementary Table 1), to yield final concentrations of 1.67 μM protein and 66.67 nM substrate in a 7.5 μl reaction. Complexes were collected at 1 min, 2.5 min, 5 min, 15 min, 45 min, 90 min, 150 min and 24 h. Cleavage was stopped by addition of 2 μl of 250 mM EDTA, 0.5% SDS and 20 μg of proteinase K. Formamide was added to the reactions with final concentration of 50%, samples were incubated at 95 °C for 10 min, and resolved on a 15% denaturing PAGE gel containing 7 M urea and imaged using a Typhoon FLA 9500 gel imager.

### Statistics and reproducibility

Nuclease activity rate constants ($k_{obs}$) were extracted from single exponential fits: $[Product] = A \times (1 - \exp(-k_{obs} \times t))$. $k_{obs}$ data are presented as mean ± s.e.m. ($n = 4$ independent replicates), obtained by direct fitting of four time-course datasets in GraphPad Prism 9 without calculating individual $k_{obs}$ values. Statistical analysis was performed using a two-sided *t*-test. The confidence interval used was 95%.

### Crystallization and X-ray structure determination

The 10-nt complementary ternary complex of dCas9 was assembled by first incubating dCas9 with the sgRNA in a 1:1.5 molar ratio, and pre-purifying the binary complex on a Superdex 200 16/600 gel filtration column (GE Healthcare) in 20 mM HEPES-KOH pH 7.5, 500 mM KCl, 1 mM DTT. The binary complex was diluted in 20 mM HEPES-KOH pH 7.5, 250 mM KCl, 1 mM DTT to 2.5 mg ml$^{-1}$ and the partially complementary dsDNA substrate was added in 1:1.5 molar excess. For crystallization, 1 μl of the ternary complex (1.5–2.5 mg ml$^{-1}$) was mixed with 1 μl of the reservoir solution (0.1 M sodium cacodylate pH 6.5, 0.8–1.2 M ammonium formate, 12–14% PEG4000) and crystals were grown at 20 °C using the hanging drop vapour diffusion setup. Crystals were collected after 3–4 weeks, cryoprotected in 0.1 M Na cacodylate pH 6.5, 1.0 M ammonium formate, 13% PEG4000, 20% glycerol, 2 mM MgCl$_2$, and flash-cooled in liquid nitrogen. Diffraction data was measured at the beamline PXIII of the Swiss Light Source at a temperature of 100 K (Paul Scherrer Institute, Villigen, Switzerland) and processed using the autoPROC and STARANISO package with anisotropic cut-off[47]. Phases were obtained by molecular replacement using the Phaser module of the Phenix package[48] using the NUC lobe of the PDB ID: 5FQ5 as initial search model. The crystals belonged to the P1 space group and contained two copies of the complex in the asymmetric unit.

### Cryo-EM sample preparation and data acquisition

To assemble the 6-, 8-, 10-, 12-, 14- and 16-nt match complexes, dCas9 protein was mixed with the sgRNA in a 1:1.5 molar ratio, and incubated at room temperature for 10 min in buffer 20 mM HEPES-KOH pH 7.5, 250 mM KCl, 1 mM DTT. The respective partially complementary dsDNA substrate (Supplementary Table 1) was then added in a 1:3 Cas9:DNA molar ratio and incubated another 20 min at room temperature. The complexes were then purified using a Superdex 200 Increase 10/300 GL gel filtration column (GE Healthcare) and eluted in 20 mM HEPES-KOH pH 7.5, 250 mM KCl, 1 mM DTT. Concentration of the monomeric peak was determined using the Qubit 4 Fluorometer Protein Assay, and then diluted to 0.275 mg ml$^{-1}$ in 20 mM HEPES-KOH pH 7.5, 250 mM KCl cold buffer. 3 μl of diluted complexes were applied to a glow discharged 200-mesh holey carbon grid (Au 1.2/1.3 Quantifoil Micro Tools), blotted for 1.5–2.5 s at 90% humidity, 20 °C, plunge frozen in liquid propane/ethane mix (Vitrobot, FEI) and stored in liquid nitrogen. To prepare the 18-nt match (checkpoint), wild-type Cas9–sgRNA complex was reconstituted with substrate DNA in 20 mM HEPES-KOH pH 7.5, 150 mM KCl, 1 mM DTT buffer, and incubated with 1 mM MgCl$_2$ for 1 min at 37 °C prior to vitrification. The 18-nt match catalytic complex was reconstituted in 20 mM HEPES-KOH pH 7.5, 100 mM KCl, 1 mM DTT buffer, and incubated with 10 mM MgCl$_2$ for 1 min at 37 °C prior to vitrification. Data collection was performed on a 300 kV FEI Titan Krios G3i microscope equipped with a Gatan Quantum Energy Filter and a K3 direct detection camera in super-resolution mode. Micrographs were recorded at a calibrated magnification of 130,000× with a pixel size of 0.325 Å and subsequently binned to 0.65 Å. Data acquisition was performed automatically using EPU with three shots per hole at −0.8 μm to −2.2 μm defocus. Data for the 18-nt match (checkpoint) complex was collected using a Titan Krios G4 equipped with a SelectrisX energy filter and a FalconIV detector at a magnification of 270,000×, pixel size of 0.45 Å, defocus −0.8 μm to −1.5 μm.

## Cryo-EM data processing

Acquired super-resolution cryo-EM data was processed using cryoSPARC[49]. Gain-corrected micrographs were imported and binned to a pixel size of 0.65 Å during patch motion correction. After patch CTF estimation, micrographs with a resolution estimation worse than 5 Å and full-frame motion distance larger than 100 Å were discarded. Initial particles were picked using blob picker with 100–140 Å particle size. Particle picks were inspected and particles with NCC scores below 0.4 were discarded. Remaining particles were extracted with a box size of 384 × 384 pixels, down-sampled to 192 × 192 pixels. After 2D classification, templates were generated using good classes and particle picking was repeated using the template picker. Duplicate particles were removed, and 2D classified Cas9 particles were used for ab initio 3D reconstruction. All partially bound complexes displayed several conformational states. After several rounds of 3D classification, classes with most detailed features were reextracted using full 384 × 384 pixel box size and subjected to non-uniform refinement to generate high-resolution reconstructions[50]. The 18-nt match (checkpoint) complex was extracted with a box size of 504 × 504 pixels. Each map was sharpened using the appropriate B-factor value to enhance structural features, and local resolution was calculated and visualized using ChimeraX[51].

## Structural model building, refinement and analysis

Manual Cas9 domain placement based on PDB model 5FQ5, model adjustment and nucleic acid building was completed using COOT[52]. Atomic model refinement was performed using Phenix.refine for X-ray data and Phenix.real_space_refine for cryo-EM[48]. The quality of refined models was assessed using MolProbity[53]. Protein-nucleic acid interactions were analysed using the PISA web server[54]. Characterization of the guide–protospacer duplex was performed using the 3DNA 2.0 web server[55]. Structural figures were generated using ChimeraX[51].

## Protein sequence alignment

Protein sequences of Cas9 orthologues harbouring the REC2 domain were obtained from UniProt. Sequence alignment was performed using MUSCLE with default parameters[56]. Alignment was visualized using Jalview with highlighting only the conservation of charged residues[57].

## Reporting summary

Further information on research design is available in the Nature Research Reporting Summary linked to this article.

## Data availability

Atomic coordinates, maps and structure factors of the reported X-ray and cryo-EM structures have been deposited in the Protein Data Bank under accession numbers 7Z4D (10-nt match complex, X-ray), 7Z4C (6-nt match complex, cryo-EM), 7Z4E (8-nt match complex, cryo-EM), 7Z4K (10-nt match complex, cryo-EM), 7Z4G (12-nt match complex, cryo-EM), 7Z4H (14-nt match complex, cryo-EM), 7Z4I (16-nt match complex, cryo-EM), 7Z4L (18-nt match checkpoint complex, cryo-EM) and 7Z4J (18-nt match catalytic complex, cryo-EM) and in the Electron Microscopy Data Bank under accession codes EMD-14493 (6-nt match complex, cryo-EM), 14494 (8-nt match complex, cryo-EM), 14500 (10-nt match complex, cryo-EM), 14496 (12-nt match complex, cryo-EM), 14497 (14-nt match complex, cryo-EM), 14498 (16-nt match complex, cryo-EM), 14501 (18-nt match checkpoint complex, cryo-EM) and 14499 (18-nt match catalytic complex, cryo-EM). Source data are provided with this paper.

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

**Acknowledgements** This work was supported by the Swiss National Science Foundation Grant 31003A_182567 (to M.J.). M.J. is an International Research Scholar of the Howard Hughes Medical Institute and Vallee Scholar of the Bert L. and N. Kuggie Vallee Foundation. We thank S. Sorrentino and A. Myasnikov for their assistance with cryogenic electron microscopy data collection; F. Boneberg and C. Chanez for their help with preparing reagents; members of the Jinek laboratory for discussion and critical reading of the manuscript; and J. Cofsky, K. Soczek and J. Doudna for sharing unpublished data and helpful comments.

**Author contributions** M.P. and M.J. conceived the study and designed experiments. M.P. purified Cas9, performed in vitro cleavage assays, crystallized 10-bp heteroduplex complex, prepared cryo-EM samples and solved the structures. I.Q., L.L. and L.M.M. expressed and purified Cas9 mutant proteins, assisted with figure preparation, and collected cryo-EM data. M.S. collected cryo-EM data. M.P. and M.J. performed structural analysis and wrote the manuscript with input from the remaining authors.

**Competing interests** The authors declare no competing interests.

**Additional information**
**Correspondence and requests for materials** should be addressed to Martin Jinek.

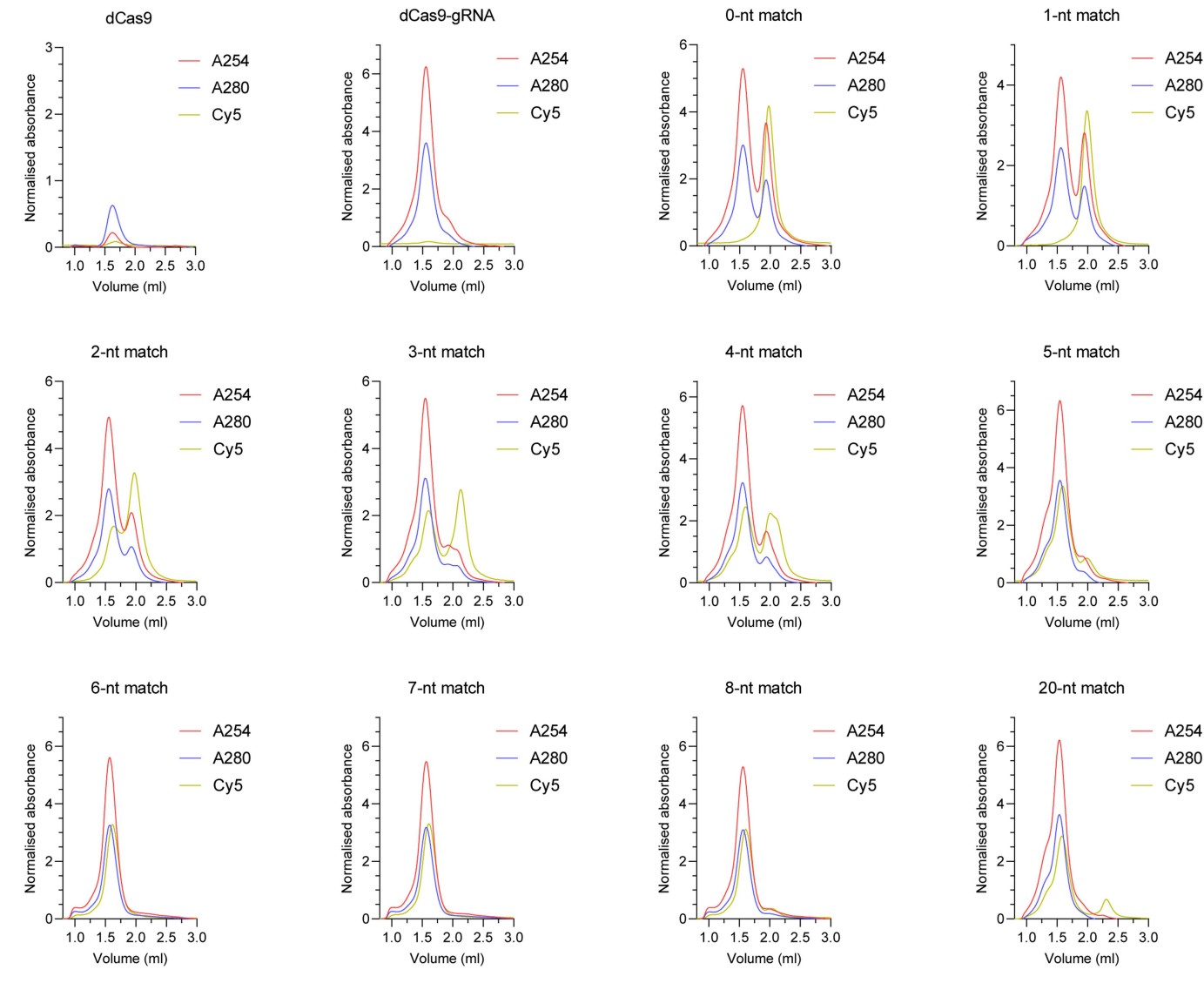

**Extended Data Fig. 1 | Minimal target complementarity necessary for stable Cas9 binding.** Size exclusion chromatography analysis of nuclease-inactive SpCas9 complexed with sgRNA and Cy5-labeled DNA substrates with increasing extent of guide-target complementarity. A254/A280 and Cy5 signals were normalised.

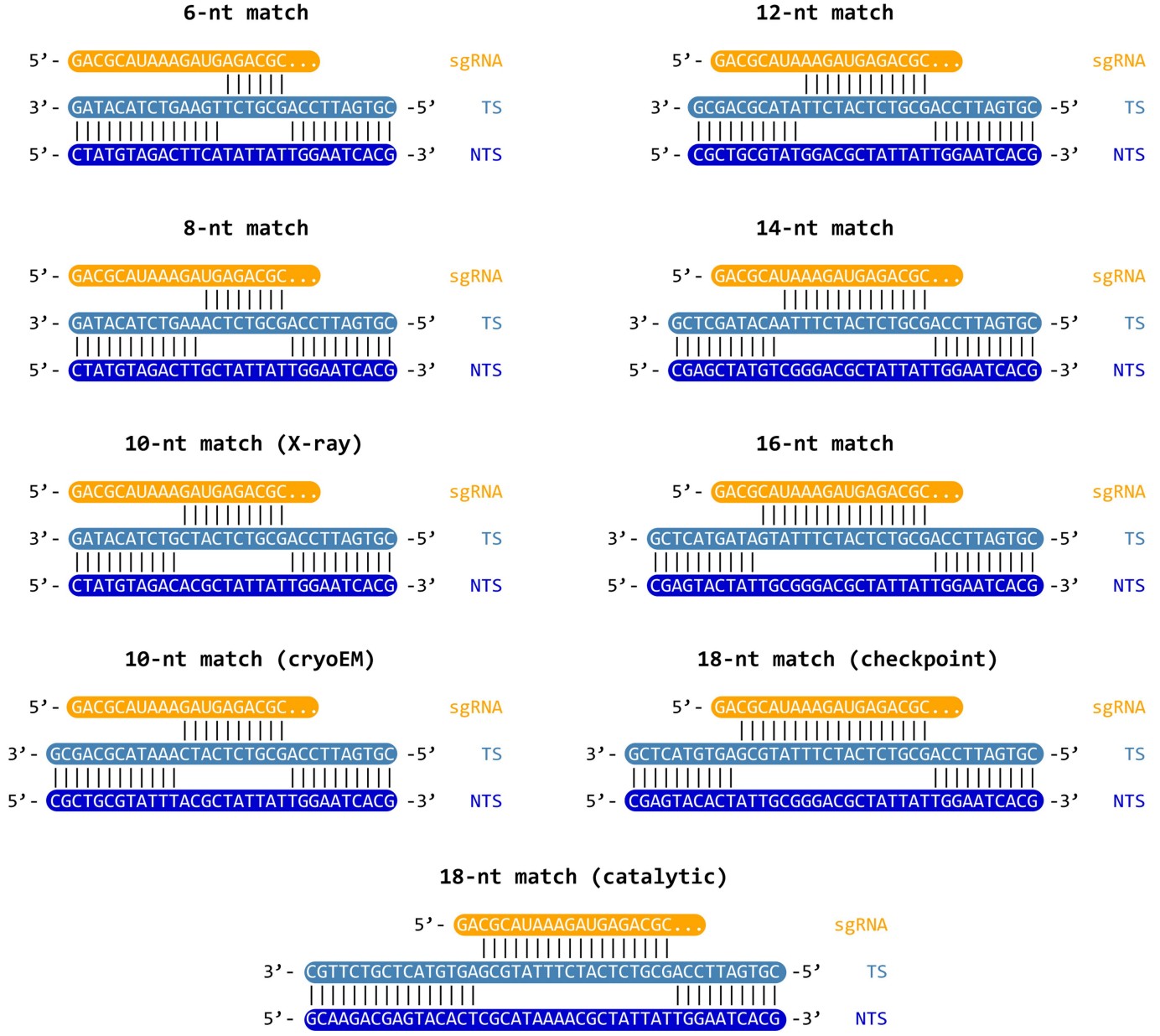

**Extended Data Fig. 2 | Schematic representation of DNA substrates used in structural studies.** Base pair complementarity between sgRNA, target strand (TS), and non-target strand (NTS) is indicated by black lines.

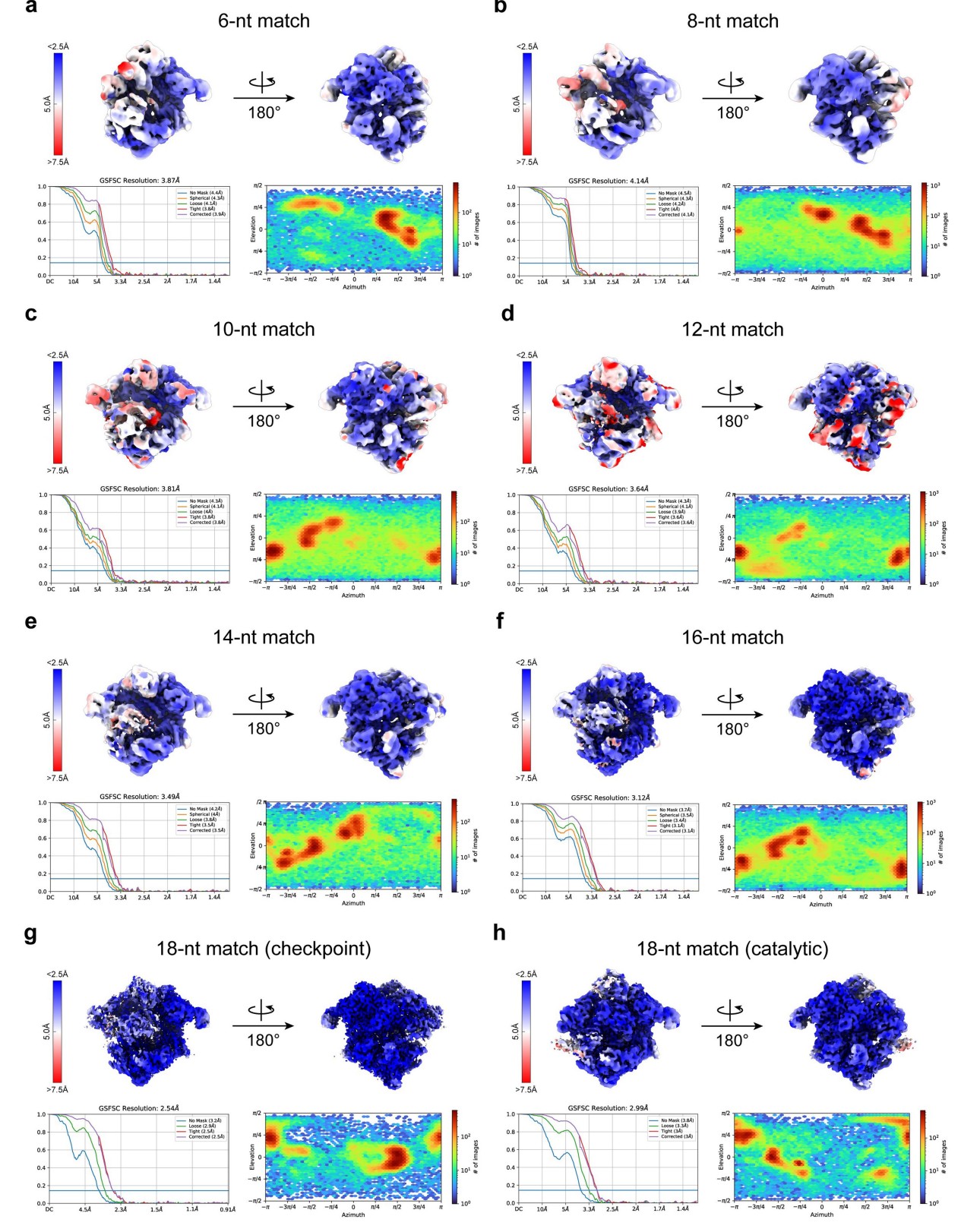

**Extended Data Fig. 3 | Cryo-EM density maps of DNA-bound SpCas9 complexes.** Front (left) and back (right) views of unsharpened cryo-EM density maps of the partially-bound SpCas9 complexes. Maps are coloured by local resolution, and gold-standard FSC of 0.143 resolution graphs and particle distribution heatmaps are indicated for each complex.

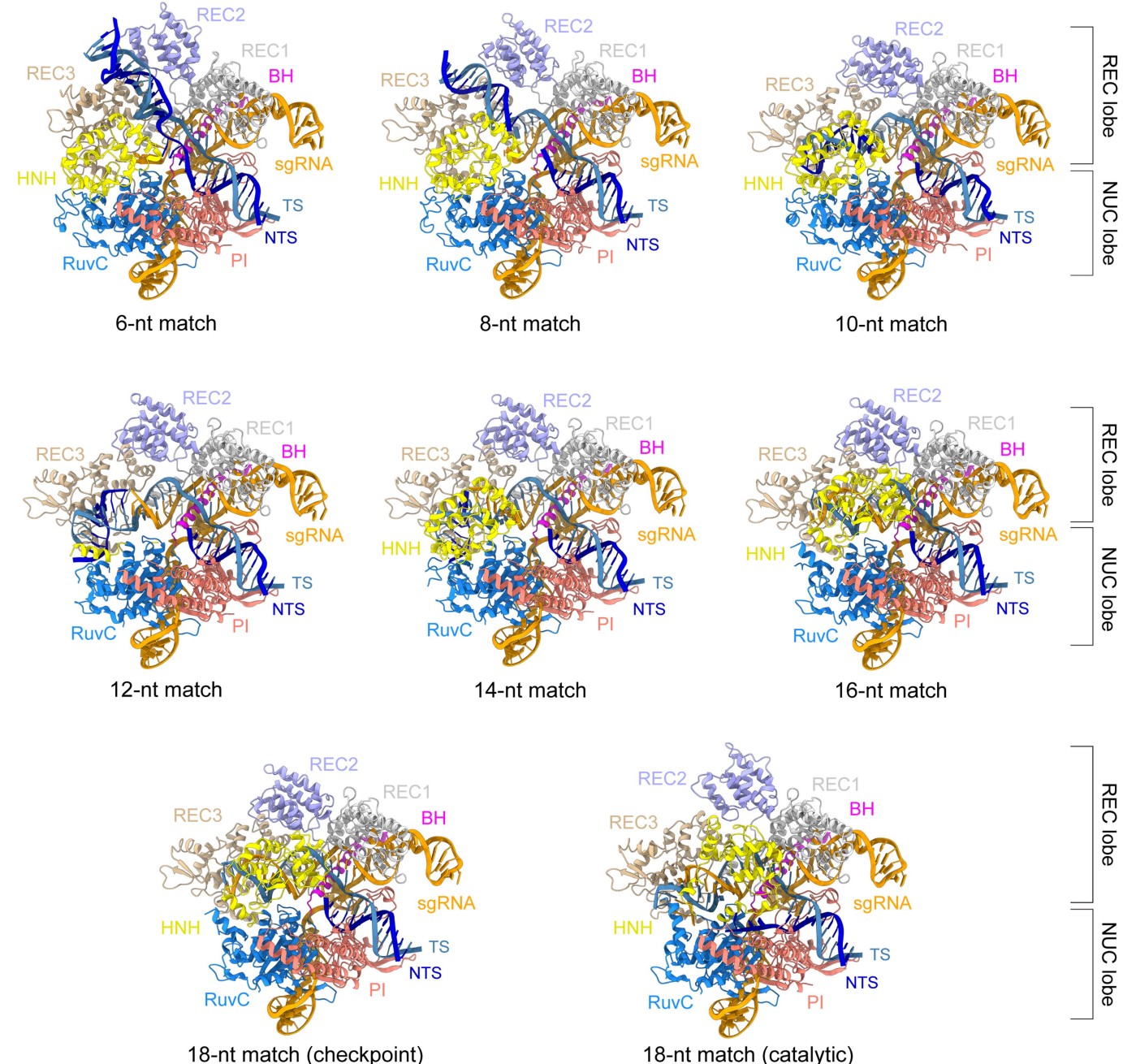

**Extended Data Fig. 4 | Structural models of DNA-bound SpCas9 complexes.** Cartoon representations of DNA-bound 6-, 8-, 10-, 12-, 14-, 16-, 18-nt match (checkpoint and catalytic) complexes of SpCas9. Each model was generated based on the corresponding map shown in Extended Data Fig. 3.

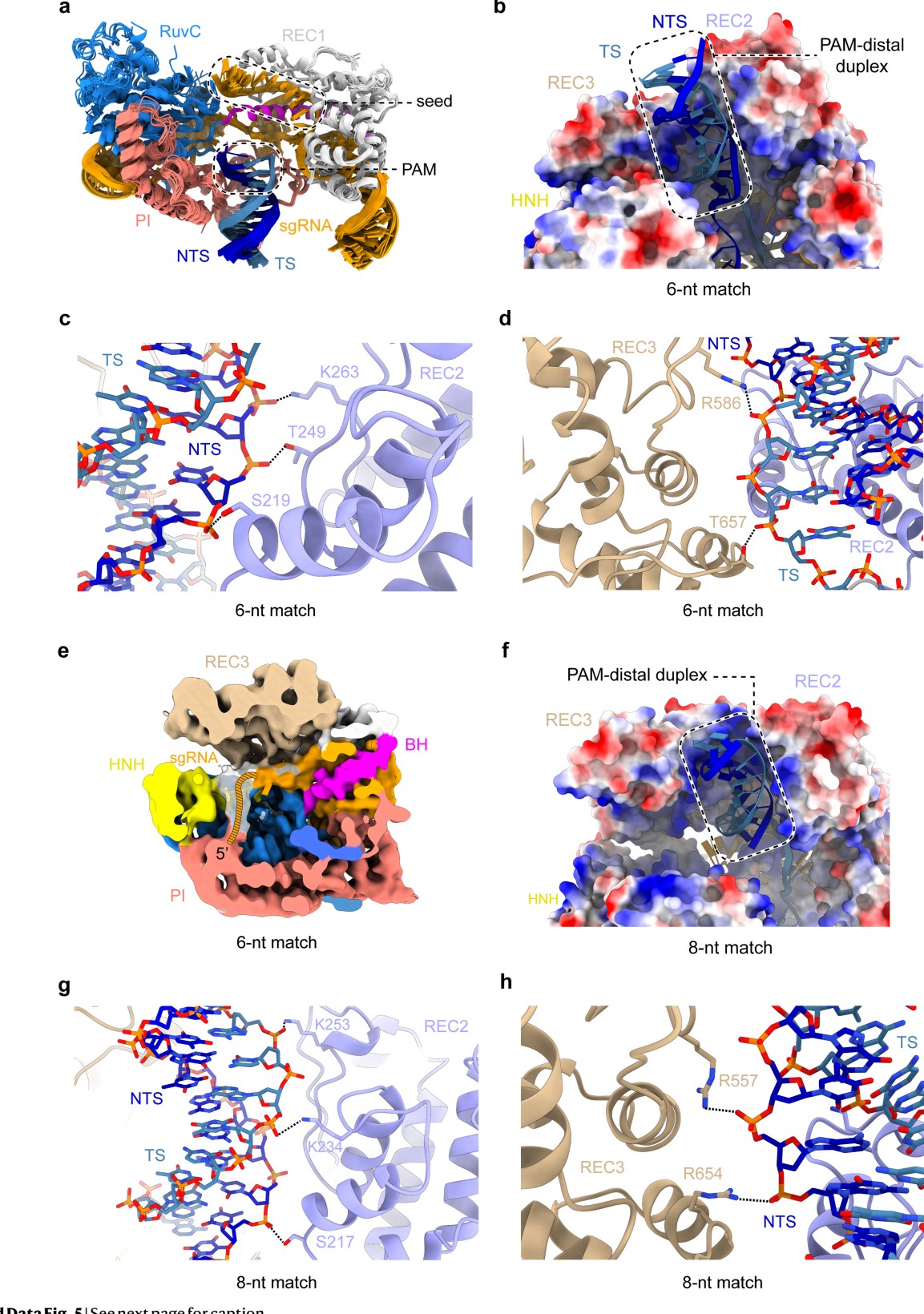

**Extended Data Fig. 5** | See next page for caption.

**Extended Data Fig. 5 | Stabilisation of the PAM-distal duplex by REC2/3 domains. a**, Structural overlays of the SpCas9 bridge helix (BH), REC1, RuvC, and PAM-interacting (PI) domains, as well as the PAM-proximal DNA duplex and the sgRNA from the partially bound complex structures determined by crystallography and cryoEM, and full R-loop complexes (PDB: 6O0X, 6O0Y, 6O0Z)[28]. **b**, Zoom-in view of the PAM-distal DNA duplex in the 6-nt match complex. The protein surface is coloured according to electrostatic surface potential, with red denoting negative and blue positive charge. **c**, Interactions between SpCas9 REC2 domain and the backbone of the PAM-distal NTS in the 6-nt match complex. **d**, Interactions between the REC3 domain and the backbone of the PAM-distal TS in the 6-nt match complex. **e**, Central slice through the 6-nt match complex. Cryo-EM density map is coloured according to Fig. 1a. White density indicates positioning of the 5′ sgRNA end. **f**, PAM-distal DNA duplex in the 8-nt match complex remains positioned in a positively charged cavity between the REC2 and REC3 domains. The protein surface is coloured according to electrostatic surface potential. **g**, Interactions of the REC2 domain with the PAM-distal DNA duplex in the 8-nt match complex. **h**, Interactions of the REC3 domain with the NTS of the PAM-distal duplex in the 8-nt match complex.

**a**

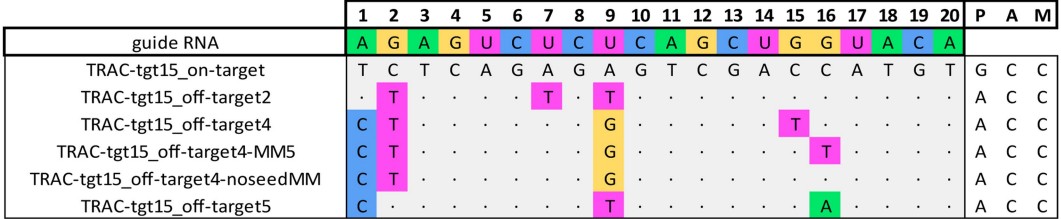

| | 1 | 2 | 3 | 4 | 5 | 6 | 7 | 8 | 9 | 10 | 11 | 12 | 13 | 14 | 15 | 16 | 17 | 18 | 19 | 20 | P | A | M |
|---|---|---|---|---|---|---|---|---|---|---|---|---|---|---|---|---|---|---|---|---|---|---|---|
| guide RNA | A | G | A | G | U | C | U | C | U | C | A | G | C | U | G | G | U | A | C | A | | | |
| TRAC-tgt15_on-target | T | C | T | C | A | G | A | G | A | G | T | C | G | A | C | C | A | T | G | T | G | C | C |
| TRAC-tgt15_off-target2 | · | T | · | · | · | · | T | · | T | · | · | · | · | · | · | · | · | · | · | · | A | C | C |
| TRAC-tgt15_off-target4 | C | T | · | · | · | · | · | · | G | · | · | · | · | · | T | · | · | · | · | · | A | C | C |
| TRAC-tgt15_off-target4-MM5 | C | T | · | · | · | · | · | · | G | · | · | · | · | · | · | T | · | · | · | · | A | C | C |
| TRAC-tgt15_off-target4-noseedMM | C | T | · | · | · | · | · | · | G | · | · | · | · | · | · | · | · | · | · | · | A | C | C |
| TRAC-tgt15_off-target5 | C | · | · | · | · | · | · | · | T | · | · | · | · | · | · | A | · | · | · | · | A | C | C |

**b**

Legend:
- WT on-target
- Y450A on-target
- WT off-target 2
- Y450A off-target 2
- WT off-target 4
- Y450A off-target 4
- WT off-target 5
- Y450A off-target 5

**c**

TRAC-tgt5 on-target

TRAC-tgt5 off-target2

Legend:
- WT
- S219A / T249A / K263A
- R586A / T657A
- S217A / K234A / K253A
- R557A / R654A
- ΔREC2

**d**

Legend:
- WT
- S219A / T249A / K263A
- R586A / T657A
- S217A / K234A / K253A
- R557A / R654A
- ΔREC2

**e**

TRAC-tgt5 on-target

TRAC-tgt5 off-target2

Legend:
- WT
- D261A / D269A / D272A / D273A / D274A / D276A / K646A / K649A / K652A / R653A / R654A / R655A
- D261A / D269A / D272A / D273A / D274A / D276A
- K646A / K649A / K652A / R653A / R654A / R655A

**f**

Legend:
- WT
- D261A / D269A / D272A / D273A / D274A / D276A / K646A / K649A / K652A / R653A / R654A / R655A
- D261A / D269A / D272A / D273A / D274A / D276A
- K646A / K649A / K652A / R653A / R654A / R655A

**Extended Data Fig. 6** | See next page for caption.

**Extended Data Fig. 6 | In vitro cleavage activities of structure-guided REC2 and REC3 mutants of Cas9. a**, Off-target sequences selected for nuclease activity assays. Nucleotide mismatches between the TRAC guide RNA and the target are highlighted; matching nucleotides are denoted by a dot. **b**, In vitro cleavage kinetics of Y450A mutants from which $k_{obs}$ values are derived using single exponential fitting. Data represents mean ± SEM (n = 4). **c**, In vitro cleavage kinetics of REC2/REC3 mutants from which $k_{obs}$ values are derived using single exponential fitting. Data represents mean ± SEM (n = 4). **d**, Cleavage rate constants of PAM-distal duplex stabilising REC2/REC3 mutants on on- and off-targets. Data represents mean fit ± SEM of n = 4 replicates, significance was determined by a two-tailed t-test. *, $p < 0.05$; **, $p < 0.01$; ***, $p < 0.001$; ****, $p < 0.0001$. **e**, In vitro cleavage kinetics of DDD and RRR helix mutants from which $k_{obs}$ values are derived using single exponential fitting. Data represents mean ± SEM (n = 4). **f**, Cleavage rate constants of Cas9 DDD and RRR helix mutants. Data represents mean fit ± SEM of n = 4 replicates, significance was determined by a two-tailed t-test. *, $p < 0.05$; **, $p < 0.01$; ***, $p < 0.001$; ****, $p < 0.0001$. On- and off-target substrates were fluorescently labelled on the PAM-proximal end of the target DNA strand in all panels.

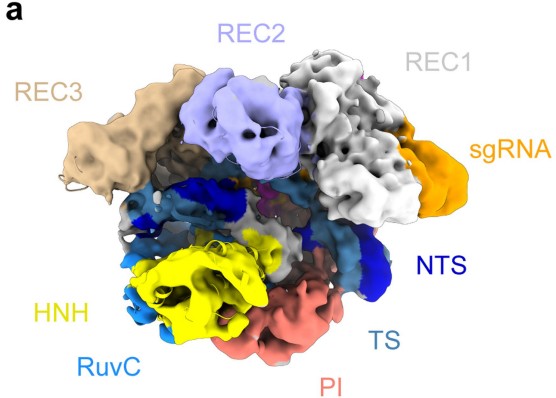

**Extended Data Fig. 7 | PAM-distal positioning and REC lobe conformation in the 10-nt match complex. a**, Cryo-EM density of the 10-nt match complex overlaid with the structural model. NTS density can be traced along the heteroduplex (white). **b**, Cartoon representations of the X-ray crystallographic structures of the 10-nt match complex as based on the two complex copies (molecules A and B) in the crystallographic asymmetric unit. The complexes exhibit highly similar conformations (RMSD 0.46 Å). **c**, Alignment of protein sequences of the REC2 DDD and REC3 RRR helices from REC2-containing Cas9 orthologs.

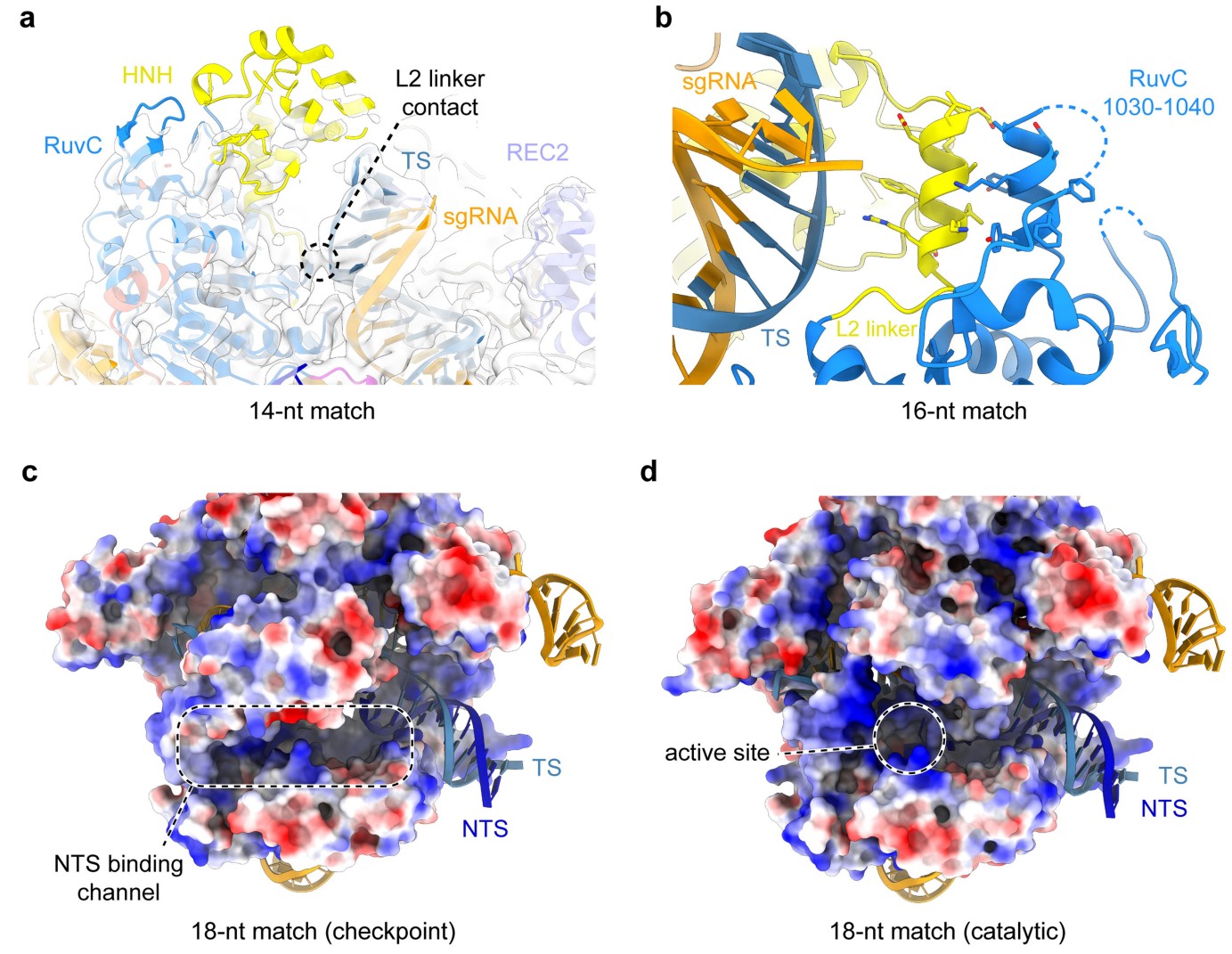

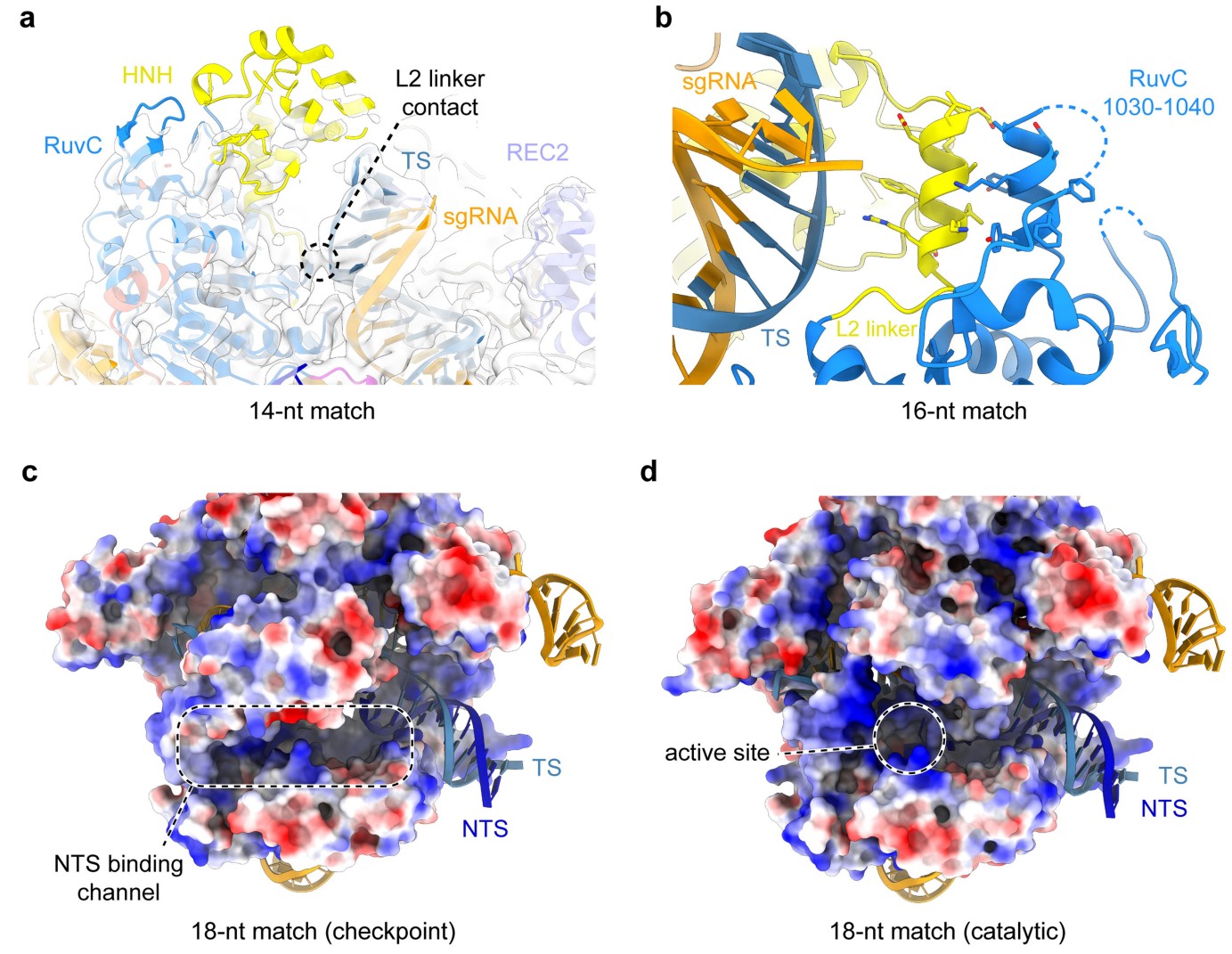 a **14-nt match**

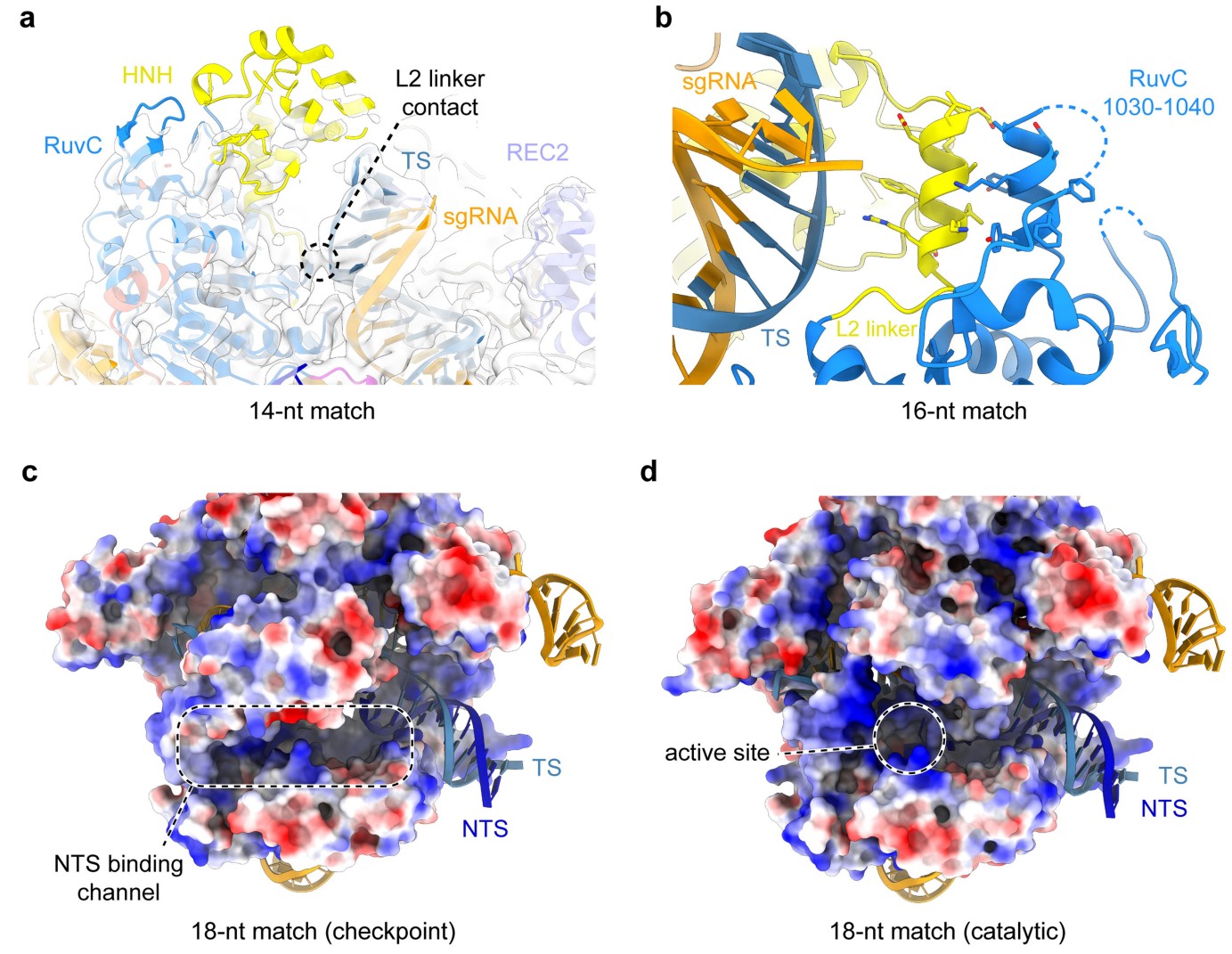 b **16-nt match**

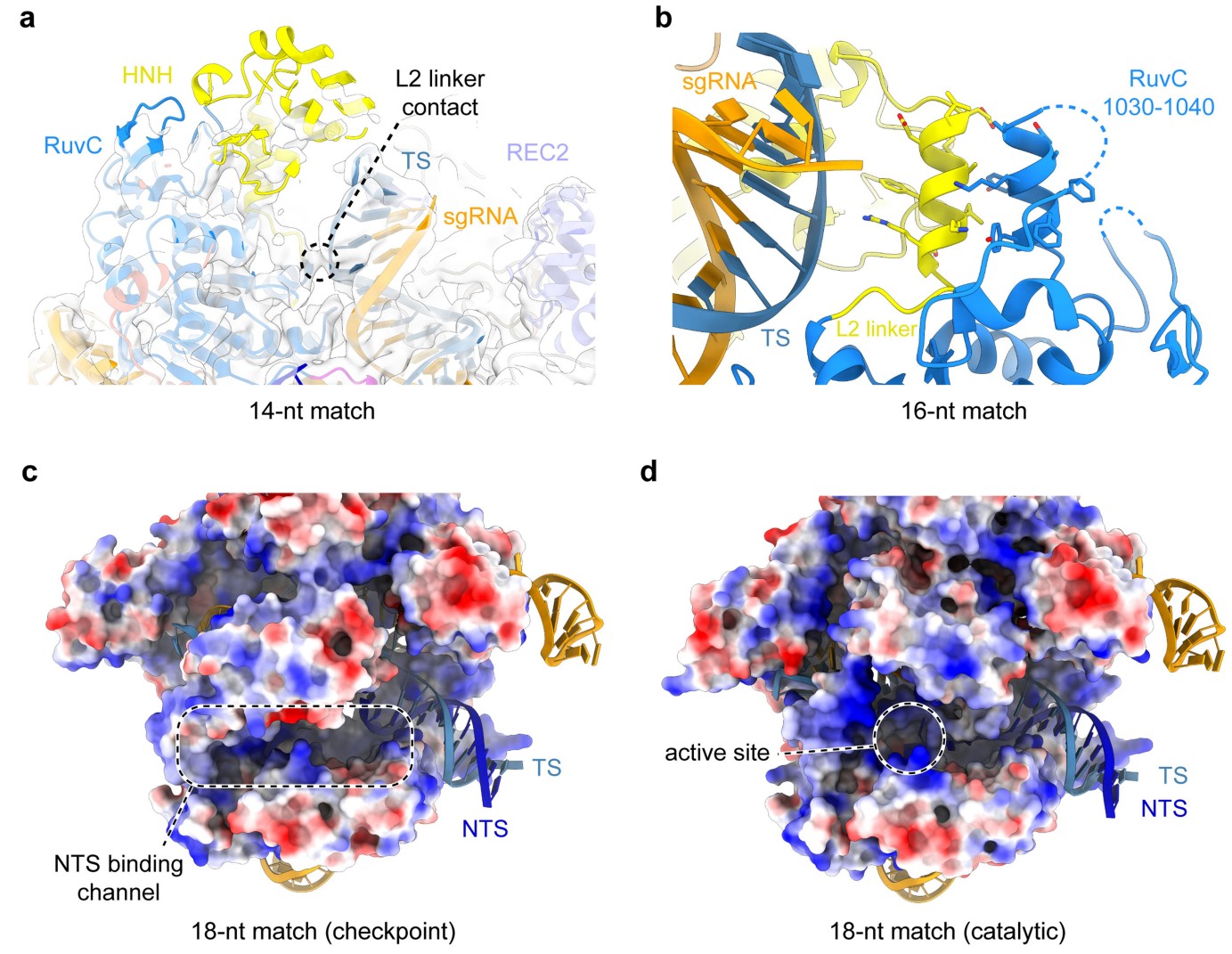 c **18-nt match (checkpoint)**

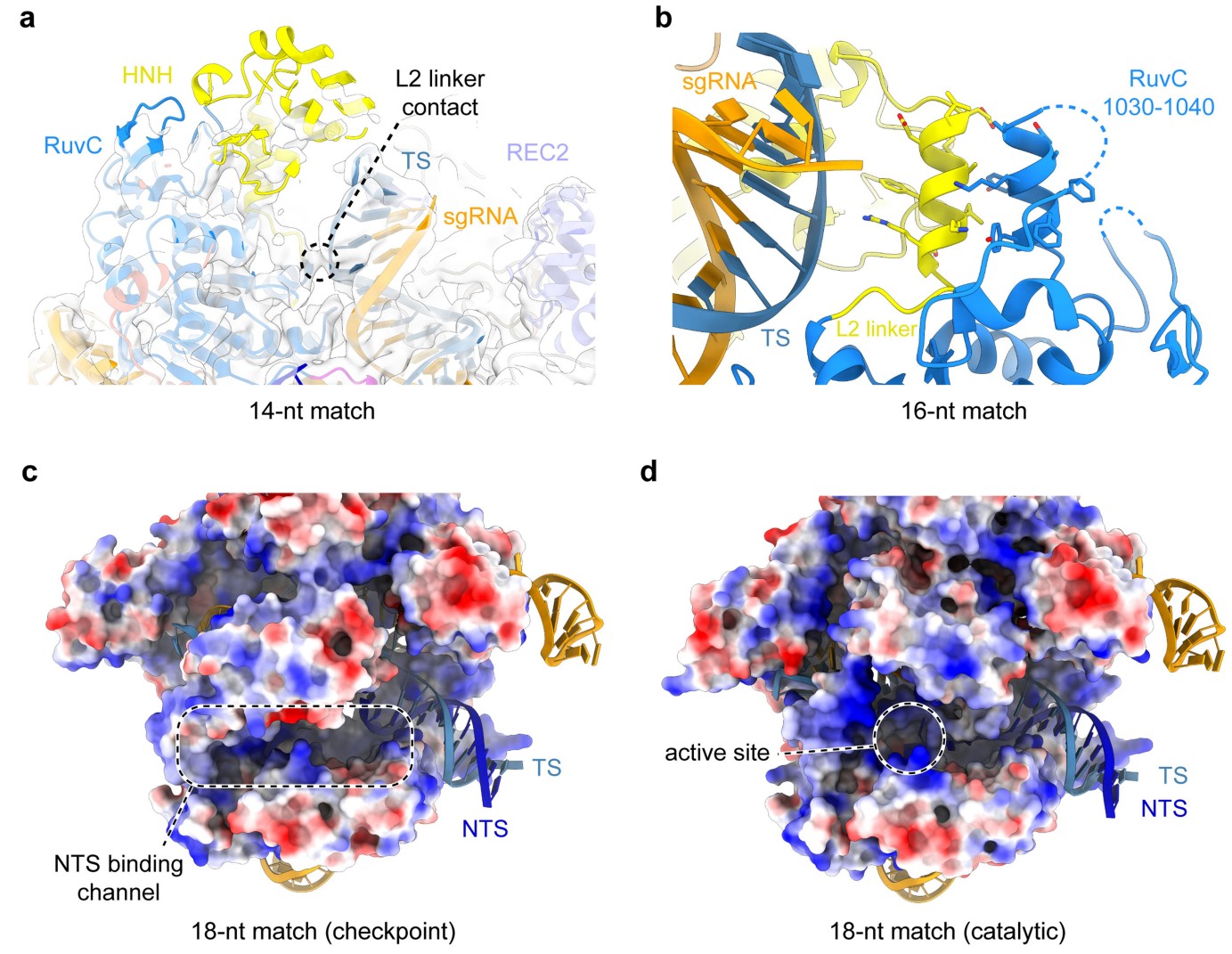 d **18-nt match (catalytic)**

**Extended Data Fig. 8 | HNH undocking induced by R-loop extension.**
**a**, Residual HNH domain density (white) observed in the 14-nt match complex, in which the elongated heteroduplex establishes a contact with the L2 linker. No NTS density is observed past the PAM region due to disorder. **b**, Zoom-in view of the interaction between the HNH domain L2 linker and the RuvC 1030–1040 helix induced by heteroduplex proximity of the 16-bp complex. **c**, HNH domain relocation towards the binding channel results in the formation of a positively charged NTS binding channel. No residual electron density (white) is observed for the NTS in the absence of the PAM-distal duplex. The protein is coloured according to electrostatic surface potential, with red being negative, blue positive. **d**, Surface electrostatics map of the 18-nt match catalytic state of SpCas9, showing the NTS binding cleft with cleaved NTS positioned within the active site.

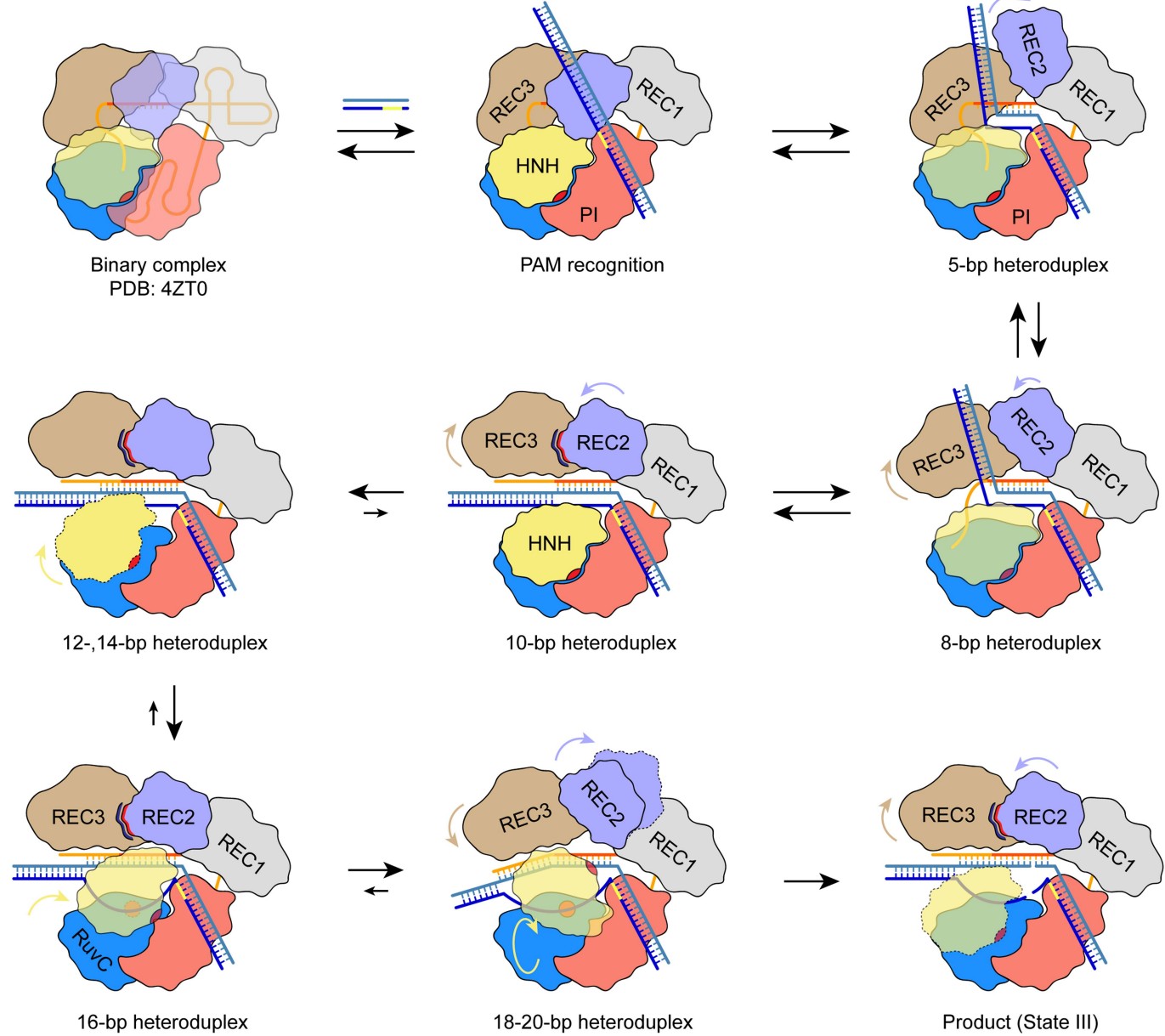

**Extended Data Fig. 9 | Molecular mechanism of Cas9 R-loop formation and conformational activation.** In the RNA-bound (binary) complex, the central DNA binding channel is occluded by the REC2 and REC3 domains. Upon PAM recognition and initial 5-nt base pairing with the seed sequence of the guide RNA, the REC2 domain is displaced to form a binding cleft to accommodate the PAM-distal DNA duplex. Formation of 8-bp heteroduplex further displaces the REC3 domain and fully opens the central binding channel, while the PAM-distal duplex remains in the REC2/3 cavity. Extension of the R-loop to 10-bp heteroduplex places the guide-TS heteroduplex and the PAM-distal duplex into the central binding channel, accompanied by formation of electrostatic contacts between the REC2 and REC3 domains. Base pairing past the seed region results in undocking of the HNH domain from the RuvC and PI domain interface, and results in its repositioning towards target heteroduplex into the checkpoint state. R-loop formation past 17 base pairs induces REC2 domain displacement from the binding channel and rotation of the HNH domain active site towards the TS cleavage site, while simultaneously positioning the NTS in the RuvC domain active site.

**Extended Data Table 1 | Cryo-EM data collection, refinement and validation statistics for 6-nt, 8-nt, 10-nt, and 12-nt match complexes**

| | dCas9-sgRNA (3-stem)-6-nt match (PDB: 7Z4C, EMDB: 14493) | dCas9-sgRNA (3-stem)-8-nt match (PDB: 7Z4E, EMDB: 14494) | dCas9-sgRNA (3-stem)-10-nt match (PDB: 7Z4K, EMDB: 14500) | dCas9-sgRNA (3-stem)-12-nt match (PDB: 7Z4G, EMDB: 14496) |
|---|---|---|---|---|
| **Data collection and processing** | | | | |
| Magnification | 130,000 | 130,000 | 130,000 | 130,000 |
| Voltage (kV) | 300 | 300 | 300 | 300 |
| Electron exposure (e–/Å$^2$) | 64.23 | 64.23 | 66.22 | 66.60 |
| Defocus range (μm) | -1.0 to -2.4 | -1.0 to -2.4 | -1.0 to -2.4 | -1.0 to -2.4 |
| Pixel size (Å) | 0.65 | 0.65 | 0.65 | 0.65 |
| Symmetry imposed | C1 | C1 | C1 | C1 |
| Initial particle images (no.) | 137,838 | 196,733 | 128,167 | 105,145 |
| Final particle images (no.) | 76,715 | 147,231 | 127,523 | 104,580 |
| Map resolution (Å) | 3.87 | 4.14 | 3.81 | 3.64 |
| FSC threshold | 0.143 | 0.143 | 0.143 | 0.143 |
| Map resolution range (Å) | 2.5-7.5 | 2.5-7.5 | 2.5-7.5 | 2.5-7.5 |
| | | | | |
| **Refinement** | | | | |
| Initial model used (PDB code) | 5FQ5 | 5FQ5 | 5FQ5 | 5FQ5 |
| Model resolution (Å) | 4.2 | 4.5 | 4.3 | 3.9 |
| FSC threshold | 0.5 | 0.5 | 0.5 | 0.5 |
| Map sharpening $B$ factor (Å$^2$) | -164.1 | -208.6 | -141.8 | -122.8 |
| Model composition | | | | |
| Non-hydrogen atoms | 14180 | 14069 | 14005 | 12714 |
| Protein residues | 1357 | 1357 | 1360 | 1188 |
| Nucleotide residues | 148 | 142 | 138 | 144 |
| Ligands | 0 | 0 | 0 | 0 |
| $B$ factors (mean, Å$^2$) | | | | |
| Protein | 239.2 | 274.9 | 256.3 | 204.9 |
| Nucleotide | 190.1 | 212.3 | 196.6 | 187.3 |
| Ligand | n.a. | n.a. | n.a. | n.a. |
| R.m.s. deviations | | | | |
| Bond lengths (Å) | 0.003 | 0.003 | 0.003 | 0.003 |
| Bond angles (°) | 0.451 | 0.546 | 0.436 | 0.478 |
| Validation | | | | |
| MolProbity score | 1.78 | 1.67 | 1.58 | 1.63 |
| Clashscore | 8.05 | 9.63 | 9.47 | 12.71 |
| Poor rotamers (%) | 2.06 | 1.64 | 0.99 | 1.03 |
| Ramachandran plot | | | | |
| Favored (%) | 97.5 | 98.0 | 97.6 | 98.1 |
| Allowed (%) | 2.5 | 2.0 | 2.4 | 1.9 |
| Disallowed (%) | 0.0 | 0.0 | 0.0 | 0.0 |

**Extended Data Table 2 | Cryo-EM data collection, refinement and validation statistics for 14-nt, 16-nt, 18-nt checkpoint, and 18-nt catalytic match complexes**

| | dCas9-sgRNA (3-stem)-14-nt match (PDB: 7Z4H, EMDB: 14497) | dCas9-sgRNA (3-stem)-16-nt match (PDB: 7Z4I, EMDB: 14498) | Cas9-sgRNA (3-stem)-18-nt match (checkpoint) (PDB: 7Z4L, EMDB: 14501) | Cas9-sgRNA (3-stem)-18-nt match (catalytic) (PDB: 7Z4J, EMDB: 14499) |
|---|---|---|---|---|
| **Data collection and processing** | | | | |
| Magnification | 130,000 | 130,000 | 270,000 | 130,000 |
| Voltage (kV) | 300 | 300 | 300 | 300 |
| Electron exposure (e–/Å²) | 67.44 | 67.24 | 60.00 | 64.20 |
| Defocus range (μm) | -1.0 to -2.4 | -1.0 to -2.4 | -0.8 to -1.5 | -1.0 to -2.4 |
| Pixel size (Å) | 0.65 | 0.65 | 0.45 | 0.65 |
| Symmetry imposed | C1 | C1 | C1 | C1 |
| Initial particle images (no.) | 229,760 | 434,090 | 597,564 | 277,414 |
| Final particle images (no.) | 75,738 | 146,573 | 66,518 | 68,772 |
| Map resolution (Å) | 3.49 | 3.12 | 2.54 | 2.99 |
| FSC threshold | 0.143 | 0.143 | 0.143 | 0.143 |
| Map resolution range (Å) | 2.5-7.5 | 2.5-7.5 | 2.5-7.5 | 2.5-7.5 |
| | | | | |
| **Refinement** | | | | |
| Initial model used (PDB code) | 5FQ5 | 5FQ5 | 5FQ5 | 5FQ5 |
| Model resolution (Å) | 3.8 | 3.4 | 2.9 | 3.2 |
| FSC threshold | 0.5 | 0.5 | 0.5 | 0.5 |
| Map sharpening $B$ factor (Å²) | -117.1 | -98.9 | -48.1 | -75.8 |
| Model composition | | | | |
| Non-hydrogen atoms | 13901 | 13834 | 13822 | 13948 |
| Protein residues | 1334 | 1338 | 1337 | 1345 |
| Nucleotide residues | 143 | 137 | 136 | 142 |
| Ligands | 0 | 2 | 3 | 4 |
| $B$ factors (mean, Å²) | | | | |
| Protein | 256.7 | 201.2 | 120.8 | 140.6 |
| Nucleotide | 196.9 | 138.1 | 78.7 | 121.4 |
| Ligand | n.a. | 149.5 | 85.6 | 81.8 |
| R.m.s. deviations | | | | |
| Bond lengths (Å) | 0.003 | 0.003 | 0.005 | 0.005 |
| Bond angles (°) | 0.452 | 0.464 | 0.525 | 0.476 |
| Validation | | | | |
| MolProbity score | 1.60 | 1.43 | 1.88 | 1.69 |
| Clashscore | 9.94 | 7.91 | 8.74 | 7.21 |
| Poor rotamers (%) | 0.92 | 1.44 | 2.83 | 2.15 |
| Ramachandran plot | | | | |
| Favored (%) | 98.0 | 98.5 | 97.7 | 97.8 |
| Allowed (%) | 2.0 | 1.5 | 2.3 | 2.2 |
| Disallowed (%) | 0.0 | 0.0 | 0.0 | 0.0 |

**Extended Data Table 3 | Crystallographic data collection and refinement statistics**

| | dCas9-sgRNA (2-stem)-10-nt match (PDB: 7Z4D) |
|---|---|
| **Data collection** | |
| Space group | P1 |
| Cell dimensions | |
| $a, b, c$ (Å) | 88.64, 95.51, 169.80 |
| $\alpha, \beta, \gamma$ (°) | 80.83, 78.04, 62.34 |
| Resolution (Å) | 66.49-3.10 (3.21 – 3.10) |
| $R_{merge}$ | 0.34 (2.18) |
| $I / \sigma I$ | 7.6 (1.3) |
| $CC_{1/2}$ | 0.988 (0.446) |
| Completeness (%) | 89.2 (56.7) |
| Multiplicity | 10.2 (10.1) |
| | |
| **Refinement** | |
| No. reflections | 85696 (4963) |
| $R_{work} / R_{free}$ | 24.7/27.5 |
| No. atoms | |
| Macromolecules | 24978 |
| Ligands | 16 |
| Solvent | 88 |
| $B$-factors | |
| Macromolecules | 69.4 |
| Ligands | 65.9 |
| Solvent | 29.7 |
| R.m.s. deviations | |
| Bond lengths (Å) | 0.003 |
| Bond angles (°) | 0.560 |

Values in parentheses are for highest-resolution shell. Structure was solved from data collected from two crystals.

# Reporting Summary

## Statistics

For all statistical analyses, confirm that the following items are present in the figure legend, table legend, main text, or Methods section.

| n/a | Confirmed | |
|---|---|---|
| ☐ | ☒ | The exact sample size ($n$) for each experimental group/condition, given as a discrete number and unit of measurement |
| ☐ | ☒ | A statement on whether measurements were taken from distinct samples or whether the same sample was measured repeatedly |
| ☐ | ☒ | The statistical test(s) used AND whether they are one- or two-sided<br>*Only common tests should be described solely by name; describe more complex techniques in the Methods section.* |
| ☒ | ☐ | A description of all covariates tested |
| ☒ | ☐ | A description of any assumptions or corrections, such as tests of normality and adjustment for multiple comparisons |
| ☐ | ☒ | A full description of the statistical parameters including central tendency (e.g. means) or other basic estimates (e.g. regression coefficient) AND variation (e.g. standard deviation) or associated estimates of uncertainty (e.g. confidence intervals) |
| ☐ | ☒ | For null hypothesis testing, the test statistic (e.g. $F$, $t$, $r$) with confidence intervals, effect sizes, degrees of freedom and $P$ value noted<br>*Give P values as exact values whenever suitable.* |
| ☒ | ☐ | For Bayesian analysis, information on the choice of priors and Markov chain Monte Carlo settings |
| ☒ | ☐ | For hierarchical and complex designs, identification of the appropriate level for tests and full reporting of outcomes |
| ☒ | ☐ | Estimates of effect sizes (e.g. Cohen's $d$, Pearson's $r$), indicating how they were calculated |

*Our web collection on statistics for biologists contains articles on many of the points above.*

## Software and code

Policy information about availability of computer code

| Data collection | EPU v2.9.0.1519REL, DA+ |
|---|---|
| Data analysis | cryoSPARC v3.2.0 (including Topaz), Phenix v1.19.1-dev4329, XDS v Mar 15, 2019 (BUILT=20190315), Coot v0.9.5, MolProbity v4.5.1, autoPROC 20210716, UCSF ChimeraX v1.3, GraphPad Prism 9, STARANISO, Jalview 2.11.1.0, MUSCLE 5, x3DNA 2.0, PISA webserver 1.48 |

For manuscripts utilizing custom algorithms or software that are central to the research but not yet described in published literature, software must be made available to editors and reviewers. We strongly encourage code deposition in a community repository (e.g. GitHub). See the Nature Portfolio guidelines for submitting code & software for further information.

## Data

Policy information about availability of data

All manuscripts must include a data availability statement. This statement should provide the following information, where applicable:

- Accession codes, unique identifiers, or web links for publicly available datasets
- A description of any restrictions on data availability
- For clinical datasets or third party data, please ensure that the statement adheres to our policy

Maps and atomic coordinates of the reported cryo-EM structures have been deposited in the Electron Microscopy Data Bank under accession codes EMDB: 14493, EMDB: 14494, EMDB: 14500, EMDB: 14496, EMDB: 14497, EMDB: 14498, EMDB: 14501, EMDB: 14499, and the Protein Data Bank with accession codes PDB: 7Z4C, PDB: 7Z4E, PDB: 7Z4K, PDB: 7Z4G, PDB: 7Z4H, PDB: 7Z4I, PDB: 7Z4L, PDB: 7Z4J. Structure factors and atomic coordinates of the reported X-ray crystallographic structure has been deposited in the Protein Data Bank with accession code PDB: 7Z4D. The model used for molecular replacement is available as PDB: 5FQ5. Cas9 sequences were obtained from Uniprot database version 2020_03.

# Field-specific reporting

Please select the one below that is the best fit for your research. If you are not sure, read the appropriate sections before making your selection.

☒ Life sciences      ☐ Behavioural & social sciences      ☐ Ecological, evolutionary & environmental sciences

For a reference copy of the document with all sections, see nature.com/documents/nr-reporting-summary-flat.pdf

# Life sciences study design

All studies must disclose on these points even when the disclosure is negative.

| | |
|---|---|
| Sample size | Sample sizes are reported in the figure legends. For nuclease activity assays, measurements of four technical replicates were performed. This would be considered standard practice for in vitro experiments of this kind. |
| Data exclusions | Particles were excluded during 2D and 3D classification during cryo-EM reconstruction. Removal of suboptimal particles is standard practice in single-particle cryoEM and is necessary to obtain homogenous and high-resolution reconstructions. |
| Replication | For nuclease activity assays, measurements of four technical replicates were performed. Information on replication of the experiments is reported in the statistics and reproducibility section in the Methods. |
| Randomization | Randomization is not applicable to this study as no live animals or human subjects were involved. |
| Blinding | Analyses performed in this manuscript were not blinded, as no live animals or human subjects were involved. Blinding is not standard practice for the presented in vitro experiments. |

# Behavioural & social sciences study design

All studies must disclose on these points even when the disclosure is negative.

| | |
|---|---|
| Study description | *Briefly describe the study type including whether data are quantitative, qualitative, or mixed-methods (e.g. qualitative cross-sectional, quantitative experimental, mixed-methods case study).* |
| Research sample | *State the research sample (e.g. Harvard university undergraduates, villagers in rural India) and provide relevant demographic information (e.g. age, sex) and indicate whether the sample is representative. Provide a rationale for the study sample chosen. For studies involving existing datasets, please describe the dataset and source.* |
| Sampling strategy | *Describe the sampling procedure (e.g. random, snowball, stratified, convenience). Describe the statistical methods that were used to predetermine sample size OR if no sample-size calculation was performed, describe how sample sizes were chosen and provide a rationale for why these sample sizes are sufficient. For qualitative data, please indicate whether data saturation was considered, and what criteria were used to decide that no further sampling was needed.* |
| Data collection | *Provide details about the data collection procedure, including the instruments or devices used to record the data (e.g. pen and paper, computer, eye tracker, video or audio equipment) whether anyone was present besides the participant(s) and the researcher, and whether the researcher was blind to experimental condition and/or the study hypothesis during data collection.* |
| Timing | *Indicate the start and stop dates of data collection. If there is a gap between collection periods, state the dates for each sample cohort.* |
| Data exclusions | *If no data were excluded from the analyses, state so OR if data were excluded, provide the exact number of exclusions and the rationale behind them, indicating whether exclusion criteria were pre-established.* |
| Non-participation | *State how many participants dropped out/declined participation and the reason(s) given OR provide response rate OR state that no participants dropped out/declined participation.* |
| Randomization | *If participants were not allocated into experimental groups, state so OR describe how participants were allocated to groups, and if allocation was not random, describe how covariates were controlled.* |

# Ecological, evolutionary & environmental sciences study design

All studies must disclose on these points even when the disclosure is negative.

| | |
|---|---|
| Study description | *Briefly describe the study. For quantitative data include treatment factors and interactions, design structure (e.g. factorial, nested, hierarchical), nature and number of experimental units and replicates.* |
| Research sample | *Describe the research sample (e.g. a group of tagged Passer domesticus, all Stenocereus thurberi within Organ Pipe Cactus National Monument), and provide a rationale for the sample choice. When relevant, describe the organism taxa, source, sex, age range and* |

| | |
|---|---|
| | *any manipulations. State what population the sample is meant to represent when applicable. For studies involving existing datasets, describe the data and its source.* |
| Sampling strategy | *Note the sampling procedure. Describe the statistical methods that were used to predetermine sample size OR if no sample-size calculation was performed, describe how sample sizes were chosen and provide a rationale for why these sample sizes are sufficient.* |
| Data collection | *Describe the data collection procedure, including who recorded the data and how.* |
| Timing and spatial scale | *Indicate the start and stop dates of data collection, noting the frequency and periodicity of sampling and providing a rationale for these choices. If there is a gap between collection periods, state the dates for each sample cohort. Specify the spatial scale from which the data are taken* |
| Data exclusions | *If no data were excluded from the analyses, state so OR if data were excluded, describe the exclusions and the rationale behind them, indicating whether exclusion criteria were pre-established.* |
| Reproducibility | *Describe the measures taken to verify the reproducibility of experimental findings. For each experiment, note whether any attempts to repeat the experiment failed OR state that all attempts to repeat the experiment were successful.* |
| Randomization | *Describe how samples/organisms/participants were allocated into groups. If allocation was not random, describe how covariates were controlled. If this is not relevant to your study, explain why.* |
| Blinding | *Describe the extent of blinding used during data acquisition and analysis. If blinding was not possible, describe why OR explain why blinding was not relevant to your study.* |

Did the study involve field work? ☐ Yes ☒ No

# Reporting for specific materials, systems and methods

We require information from authors about some types of materials, experimental systems and methods used in many studies. Here, indicate whether each material, system or method listed is relevant to your study. If you are not sure if a list item applies to your research, read the appropriate section before selecting a response.

## Materials & experimental systems

| n/a | Involved in the study |
|---|---|
| ☒ | ☐ Antibodies |
| ☒ | ☐ Eukaryotic cell lines |
| ☒ | ☐ Palaeontology and archaeology |
| ☒ | ☐ Animals and other organisms |
| ☒ | ☐ Human research participants |
| ☒ | ☐ Clinical data |
| ☒ | ☐ Dual use research of concern |

## Methods

| n/a | Involved in the study |
|---|---|
| ☒ | ☐ ChIP-seq |
| ☒ | ☐ Flow cytometry |
| ☒ | ☐ MRI-based neuroimaging |

## Antibodies

| | |
|---|---|
| Antibodies used | *Describe all antibodies used in the study; as applicable, provide supplier name, catalog number, clone name, and lot number.* |
| Validation | *Describe the validation of each primary antibody for the species and application, noting any validation statements on the manufacturer's website, relevant citations, antibody profiles in online databases, or data provided in the manuscript.* |

## Eukaryotic cell lines

Policy information about cell lines

| | |
|---|---|
| Cell line source(s) | *State the source of each cell line used.* |
| Authentication | *Describe the authentication procedures for each cell line used OR declare that none of the cell lines used were authenticated.* |
| Mycoplasma contamination | *Confirm that all cell lines tested negative for mycoplasma contamination OR describe the results of the testing for mycoplasma contamination OR declare that the cell lines were not tested for mycoplasma contamination.* |
| Commonly misidentified lines (See ICLAC register) | *Name any commonly misidentified cell lines used in the study and provide a rationale for their use.* |

## Palaeontology and Archaeology

| | |
|---|---|
| Specimen provenance | *Provide provenance information for specimens and describe permits that were obtained for the work (including the name of the issuing authority, the date of issue, and any identifying information). Permits should encompass collection and, where applicable, export.* |

| Specimen deposition | *Indicate where the specimens have been deposited to permit free access by other researchers.* |
| --- | --- |
| Dating methods | *If new dates are provided, describe how they were obtained (e.g. collection, storage, sample pretreatment and measurement), where they were obtained (i.e. lab name), the calibration program and the protocol for quality assurance OR state that no new dates are provided.* |

☐ Tick this box to confirm that the raw and calibrated dates are available in the paper or in Supplementary Information.

| Ethics oversight | *Identify the organization(s) that approved or provided guidance on the study protocol, OR state that no ethical approval or guidance was required and explain why not.* |
| --- | --- |

Note that full information on the approval of the study protocol must also be provided in the manuscript.

# Animals and other organisms

Policy information about studies involving animals; ARRIVE guidelines recommended for reporting animal research

| Laboratory animals | *For laboratory animals, report species, strain, sex and age OR state that the study did not involve laboratory animals.* |
| --- | --- |
| Wild animals | *Provide details on animals observed in or captured in the field; report species, sex and age where possible. Describe how animals were caught and transported and what happened to captive animals after the study (if killed, explain why and describe method; if released, say where and when) OR state that the study did not involve wild animals.* |
| Field-collected samples | *For laboratory work with field-collected samples, describe all relevant parameters such as housing, maintenance, temperature, photoperiod and end-of-experiment protocol OR state that the study did not involve samples collected from the field.* |
| Ethics oversight | *Identify the organization(s) that approved or provided guidance on the study protocol, OR state that no ethical approval or guidance was required and explain why not.* |

Note that full information on the approval of the study protocol must also be provided in the manuscript.

# Human research participants

Policy information about studies involving human research participants

| Population characteristics | *Describe the covariate-relevant population characteristics of the human research participants (e.g. age, gender, genotypic information, past and current diagnosis and treatment categories). If you filled out the behavioural & social sciences study design questions and have nothing to add here, write "See above."* |
| --- | --- |
| Recruitment | *Describe how participants were recruited. Outline any potential self-selection bias or other biases that may be present and how these are likely to impact results.* |
| Ethics oversight | *Identify the organization(s) that approved the study protocol.* |

Note that full information on the approval of the study protocol must also be provided in the manuscript.

# Clinical data

Policy information about clinical studies
All manuscripts should comply with the ICMJE guidelines for publication of clinical research and a completed CONSORT checklist must be included with all submissions.

| Clinical trial registration | *Provide the trial registration number from ClinicalTrials.gov or an equivalent agency.* |
| --- | --- |
| Study protocol | *Note where the full trial protocol can be accessed OR if not available, explain why.* |
| Data collection | *Describe the settings and locales of data collection, noting the time periods of recruitment and data collection.* |
| Outcomes | *Describe how you pre-defined primary and secondary outcome measures and how you assessed these measures.* |

# Dual use research of concern

Policy information about dual use research of concern

## Hazards

Could the accidental, deliberate or reckless misuse of agents or technologies generated in the work, or the application of information presented in the manuscript, pose a threat to:

No | Yes
☒ ☐ Public health
☒ ☐ National security
☒ ☐ Crops and/or livestock
☒ ☐ Ecosystems
☒ ☐ Any other significant area

## Experiments of concern

Does the work involve any of these experiments of concern:

No | Yes
☒ ☐ Demonstrate how to render a vaccine ineffective
☒ ☐ Confer resistance to therapeutically useful antibiotics or antiviral agents
☒ ☐ Enhance the virulence of a pathogen or render a nonpathogen virulent
☒ ☐ Increase transmissibility of a pathogen
☒ ☐ Alter the host range of a pathogen
☒ ☐ Enable evasion of diagnostic/detection modalities
☒ ☐ Enable the weaponization of a biological agent or toxin
☒ ☐ Any other potentially harmful combination of experiments and agents

# ChIP-seq

## Data deposition

☐ Confirm that both raw and final processed data have been deposited in a public database such as GEO.

☐ Confirm that you have deposited or provided access to graph files (e.g. BED files) for the called peaks.

**Data access links**
*May remain private before publication.*

*For "Initial submission" or "Revised version" documents, provide reviewer access links. For your "Final submission" document, provide a link to the deposited data.*

**Files in database submission**

*Provide a list of all files available in the database submission.*

**Genome browser session**
(e.g. UCSC)

*Provide a link to an anonymized genome browser session for "Initial submission" and "Revised version" documents only, to enable peer review. Write "no longer applicable" for "Final submission" documents.*

## Methodology

**Replicates**

*Describe the experimental replicates, specifying number, type and replicate agreement.*

**Sequencing depth**

*Describe the sequencing depth for each experiment, providing the total number of reads, uniquely mapped reads, length of reads and whether they were paired- or single-end.*

**Antibodies**

*Describe the antibodies used for the ChIP-seq experiments; as applicable, provide supplier name, catalog number, clone name, and lot number.*

**Peak calling parameters**

*Specify the command line program and parameters used for read mapping and peak calling, including the ChIP, control and index files used.*

**Data quality**

*Describe the methods used to ensure data quality in full detail, including how many peaks are at FDR 5% and above 5-fold enrichment.*

**Software**

*Describe the software used to collect and analyze the ChIP-seq data. For custom code that has been deposited into a community repository, provide accession details.*

# Flow Cytometry

## Plots

Confirm that:

☐ The axis labels state the marker and fluorochrome used (e.g. CD4-FITC).

☐ The axis scales are clearly visible. Include numbers along axes only for bottom left plot of group (a 'group' is an analysis of identical markers).

☐ All plots are contour plots with outliers or pseudocolor plots.

☐ A numerical value for number of cells or percentage (with statistics) is provided.

## Methodology

| | |
|---|---|
| Sample preparation | *Describe the sample preparation, detailing the biological source of the cells and any tissue processing steps used.* |
| Instrument | *Identify the instrument used for data collection, specifying make and model number.* |
| Software | *Describe the software used to collect and analyze the flow cytometry data. For custom code that has been deposited into a community repository, provide accession details.* |
| Cell population abundance | *Describe the abundance of the relevant cell populations within post-sort fractions, providing details on the purity of the samples and how it was determined.* |
| Gating strategy | *Describe the gating strategy used for all relevant experiments, specifying the preliminary FSC/SSC gates of the starting cell population, indicating where boundaries between "positive" and "negative" staining cell populations are defined.* |

☐ Tick this box to confirm that a figure exemplifying the gating strategy is provided in the Supplementary Information.

# Magnetic resonance imaging

## Experimental design

| | |
|---|---|
| Design type | *Indicate task or resting state; event-related or block design.* |
| Design specifications | *Specify the number of blocks, trials or experimental units per session and/or subject, and specify the length of each trial or block (if trials are blocked) and interval between trials.* |
| Behavioral performance measures | *State number and/or type of variables recorded (e.g. correct button press, response time) and what statistics were used to establish that the subjects were performing the task as expected (e.g. mean, range, and/or standard deviation across subjects).* |

## Acquisition

| | |
|---|---|
| Imaging type(s) | *Specify: functional, structural, diffusion, perfusion.* |
| Field strength | *Specify in Tesla* |
| Sequence & imaging parameters | *Specify the pulse sequence type (gradient echo, spin echo, etc.), imaging type (EPI, spiral, etc.), field of view, matrix size, slice thickness, orientation and TE/TR/flip angle.* |
| Area of acquisition | *State whether a whole brain scan was used OR define the area of acquisition, describing how the region was determined.* |

Diffusion MRI ☐ Used ☐ Not used

## Preprocessing

| | |
|---|---|
| Preprocessing software | *Provide detail on software version and revision number and on specific parameters (model/functions, brain extraction, segmentation, smoothing kernel size, etc.).* |
| Normalization | *If data were normalized/standardized, describe the approach(es): specify linear or non-linear and define image types used for transformation OR indicate that data were not normalized and explain rationale for lack of normalization.* |
| Normalization template | *Describe the template used for normalization/transformation, specifying subject space or group standardized space (e.g. original Talairach, MNI305, ICBM152) OR indicate that the data were not normalized.* |
| Noise and artifact removal | *Describe your procedure(s) for artifact and structured noise removal, specifying motion parameters, tissue signals and physiological signals (heart rate, respiration).* |

| Volume censoring | *Define your software and/or method and criteria for volume censoring, and state the extent of such censoring.* |

## Statistical modeling & inference

| Model type and settings | *Specify type (mass univariate, multivariate, RSA, predictive, etc.) and describe essential details of the model at the first and second levels (e.g. fixed, random or mixed effects; drift or auto-correlation).* |

| Effect(s) tested | *Define precise effect in terms of the task or stimulus conditions instead of psychological concepts and indicate whether ANOVA or factorial designs were used.* |

Specify type of analysis: ☐ Whole brain ☐ ROI-based ☐ Both

Statistic type for inference
(See Eklund et al. 2016)

*Specify voxel-wise or cluster-wise and report all relevant parameters for cluster-wise methods.*

| Correction | *Describe the type of correction and how it is obtained for multiple comparisons (e.g. FWE, FDR, permutation or Monte Carlo).* |

## Models & analysis

n/a | Involved in the study
☒ ☐ Functional and/or effective connectivity
☒ ☐ Graph analysis
☒ ☐ Multivariate modeling or predictive analysis

