## [Peer Review File · Nature]

Manuscript Title: R-loop formation and conformational activation mechanisms of Cas9

Reviewer Comments & Author Rebuttals

Reviewer Reports on the Initial Version:

Referee #1: kinetic analysis of nucleic acid processes

Referee #2: CRISPR cryoEM

Referee #3: CRISPR biology

Referees' comments:

Referee #1:

Pacesa and Jinek use cryo-EM to determine structures of dCas9 bound to a series of DNA constructs with variable lengths of complementarity, allowing the formation of R-loops of 6, 8, 10, 12, 14, or 16 bp. Although the broad contours of the conformational changes that accompany R-loop formation were known from previous structures with crRNA only or with a full 20-bp R-loop, the current work increases the understanding of the steps in this process and how the lengthening of the R-loop during its propagation induces and/or captures protein rearrangements, ultimately leading to the fully activated complex.

Specifically, there are two principal new findings. First, a rearrangement occurs in going from the potential 6-bp R-loop to the 8-bp R-loop, which includes repositioning of a Y residue that would otherwise block R-loop propagation (and indeed limits what would be a 6-bp R-loop to only 5 bp), and movement of the REC2 and REC3 domains. The authors suggest that an intermediate state previously observed by FRET corresponds to this 5-bp R-loop structure. Second, for the structures with 12-, 14-, and 16-bp R-loops, Cas9 is in an inactive conformation, with the HNH domain position and orientation being incompatible with target strand cleavage. This inactive conformation has been characterized previously by others using biophysical approaches including FRET, and large motions of the HNH domain corresponding to different states of Cas9 activation have been demonstrated in prior structural studies.

While there is improved resolution afforded by the current study in terms of the R-loop length dependence of the Cas9 conformational changes, the extent of truly new information seems a bit modest. Additionally, it is not established in the current work that the specific features of the characterized conformations represent on-pathway intermediates during R-loop propagation. In the absence of parallel biophysical studies to detect key features of the structural changes and dynamics, the possibility exists that some structural features are off-pathway conformations that

ultimately accumulate with the partial R-loops but would not form during the normal process of R-loop formation with a complementary target DNA.

The authors also suggest that the structures establish a set of amino acid residues as good targets for mutations that might increase Cas9 specificity against mismatches. These residues contact the PAM-distal DNA duplex backbone during the transition from the 5-bp R-loop to the 8-bp R-loop, and the suggestion is that mutations in them would further destabilize an intermediate state to promote dissociation of off-target sequences and improve specificity. First, it is unclear what intermediate is being referred to, because if the contacts are not formed until the R-loop is 8 bp, mutating these amino acids would not be expected to destabilize the intermediate with the shorter R-loop. Second, regardless of which intermediate is being considered, the logic of these mutations improving specificity is not clear, because the contacts are apparently formed in the context of a matched duplex, so the simplest expectation is that mutations would destabilize the matched and mismatched complexes equally, most simply with no impact on specificity. It is possible that there is a more nuanced suggestion to be made regarding the specific kinetic steps in R-loop formation that are specificity determining, but the point is not at all clear in its current form.

Referee #2:

The authors determined a series of structures of SpyCas9 in complex with a sgRNA and partially matched DNA substrates that contain 6, 8, 10, 12, 14 and 16 complementary nucleotides. The authors used incomplete complementary targets to prevent further extension of the guide-target heteroduplexes to capture different states of the Cas9 along target DNA binding. These structures show, like a movie, the conformational changes of Cas9 and guide-target heteroduplexes during target DNA binding. In particular, they revealed detailed rearrangements of the Rec2, Rec3 and HNH structural domains along their DNA binding pathways. They found that incomplete target pairing does not induce the conformational displacement required for nuclease structural domain activation, explaining why the non-target complex remains trapped in the inactive state. However, although this paper resolves the several structures of different intermediates and reveals structural rearrangement during DNA binding, it does not provide important insights into the molecular mechanisms of on/off-target discrimination.

The following are my major concerns:

1. In the abstract, the authors stated that the incomplete target strand pairing fails to induce the conformational displacements necessary for nuclease domain activation. Do all imperfectly complementary target strands fail to activate Cas9, and if not, what kind of mismatches fail to activate the HNH domain, and what is the effect of the number and position of mismatched nucleotides?
2. Mismatches between the guide and target in the seed region abolish Cas9 DNA cleavage, while mismatches in other region has less effect, resulting in off-target cleavage. The mismatches may affect R-loop formation and decrease or abolish Cas9 DNA cleavage activity. Do single or double nucleotide mismatches in the seed region abolish the activity by preventing R-loop formation? How

many nucleotides mismatches block the guide-target heteroduplex propagation? How does the mismatch in different position have different effect?

3. The TS and NTS remain hybridized at the PAM-distal end of the DNA. How many base pairs between are observed in 6-nt, 8-nt and 10-nt match structures? The authors found the relocation of the PAM-distal duplex in distinct structures. The superposition of the PAM-distal duplex is needed to show its conformational changes.

4. Page 5: The authors found that the 6-nt complex only forms a 5-bp guide-target duplex while the last base pair stacks with Y450, and in the structure of the 8-nt complex R-loop heteroduplex past Tyr450. The authors compared the structures of the 6-nt and 8-nt complexes and concluded that "These observations suggest that the seed sequence of the Cas9 guide RNA is bipartite and that its hybridization with target DNA proceeds in two step..." How do the heteroduplex lengths and mild conformational changes indicate that the seed sequence of the guide RNA is bipartite? The data here are somewhat over-interpreted.

5. In the 6-nt and 8-nt complementary complexes, both Rec2 and Rec3 form interactions with the PAM-distal duplex. For the fully complementary target DNA, do Rec2 and Rec3 also form similar interactions with the PAM-distal region during R-loop extension? That is, is there any intermediate state with the similar interaction along the target DNA binding? Are these hydrogen bonds observed in the 6-nt and 8-nt complementary complexes unique to these off-target DNAs? On p. 5, "Mutation of these residues, which would further destabilize this intermediate state and thus promote off-target dissociation, presents an opportunity to generate novel high-fidelity SpCas9 variants", the mutation data support this idea are required here.

6. In Fig. 1e (right panel) and Fig. 2a, the direction of the arrows indicating the repositioning of Rec2 is reversed, but these two figures show that the motion of Rec2 is in the same direction. Structural overlay of the REC2 and REC3 domains in the 6-, 8- and 10-nt match complexes may clarify the structural rearrangement of these domains.

7. Based on the structures in this manuscript, are the authors able to predicate the conformational changes in the Rec lobe of Cas9 lacking the REC2 domain during target binding?

Minor

1. Page 7: Ext. Data Fig. 9b is Ext. Data Fig. 8b.

2. In Ext. Data Table 3, what do the nucleotides in red indicate?

Referee #3:

Pacesa and Jinek have used a cryo-EM approach to study relevant details of R-loop formation and the coupling to conformational activation of Cas9. By a systematic, step-wise increase of the match between guide and target, intermediate stages of the Cas9 domains were obtained at an

unprecedented resolution. This is a very well executed, and very well described, story that, as the authors suggest, will provide a solid basis for future engineering of Cas9, to further enhance its specificity and possibly its activity.

Comments

1) The authors state that off-target formation generally results in enhanced dissociation (in case of PAM-proximal mismatches) or in a stably-bound, cleavage-incompatible state (PAM-distal mismatches). If that would be the complete story, then off-target interactions would not be a major problem for a range of genome editing applications. It would be appropriate if the authors state that early studies reported rather high cases of off-target indel formation, a result of cleavage and NHEJ-based repair. Moreover, the latter activities would make Cas9 a more robust defense system, that could still target potential escape viruses. Please elaborate a bit on this.

2) The observation of a bi-partite seed, with a 2-step base pairing process, reminded me of the situation in Cascade, where the seed is split in two segments of 5 and 2-5 nucleotides. Although I am aware of the major differences of the two systems, I would be curious if the authors consider this an analogous way of step-wise 'quality control'?

3) I remember in previous studies that Cas9-based kinking of dsDNA has been reported, at least by the Doudna group. Has this been observed in the structures presented here?

4) It is appreciated that the authors, apart from just stating that the provided new insights could provide a basis for optimizing the functionality of Cas9, included concrete suggestions. That comes with the risk that greedy reviewers will ask for more. So here we go: it would be great if the authors would actually test the phenotype of a mutant in which (some of) the positively-charged residues of REC2 and REC3 are substituted.

Author Rebuttals to Initial Comments:

Nature manuscript 2021-09-14915

“Mechanism of R-loop formation and conformational activation of Cas9”

Response to Referees’ Comments

Referee #1:

Pacesa and Jinek use cryo-EM to determine structures of dCas9 bound to a series of DNA constructs with variable lengths of complementarity, allowing the formation of R-loops of 6, 8, 10, 12, 14, or 16 bp. Although the broad contours of the conformational changes that accompany R-loop formation were known from previous structures with crRNA only or with a full 20-bp R-loop, the current work increases the understanding of the steps in this process and how the lengthening of the R-loop during its propagation induces and/or captures protein rearrangements, ultimately leading to the fully activated complex.

Specifically, there are two principal new findings. First, a rearrangement occurs in going from the potential 6-bp R-loop to the 8-bp R-loop, which includes repositioning of a Y residue that would otherwise block R-loop propagation (and indeed limits what would be a 6-bp R-loop to only 5 bp), and movement of the REC2 and REC3 domains. The authors suggest that an intermediate state previously observed by FRET corresponds to this 5-bp R-loop structure. Second, for the structures with 12-, 14-, and 16-bp R-loops, Cas9 is in an inactive conformation, with the HNH domain position and orientation being incompatible with target strand cleavage. This inactive conformation has been characterized previously by others using biophysical approaches including FRET, and large motions of the HNH domain corresponding to different states of Cas9 activation have been demonstrated in prior structural studies.

We thank the Referee for the positive feedback and the constructive comments.

While there is improved resolution afforded by the current study in terms of the R-loop length dependence of the Cas9 conformational changes, the extent of truly new information seems a bit modest. Additionally, it is not established in the current work that the specific features of the characterized conformations represent on-pathway intermediates during R-loop propagation. In the absence of parallel biophysical studies to detect key features of the structural changes and dynamics, the possibility exists that some structural features are off-pathway conformations that ultimately accumulate with the partial R-loops but would not form during the normal process of R-loop formation with a complementary target DNA.

We thank the Referee for raising this point. Previous biochemical and biophysical studies have clearly demonstrated that DNA binding by Cas9 is a directional process that starts with DNA duplex melting and nucleation of the guide-target heteroduplex immediately upstream of the PAM, followed by directional extension of the R-loop towards the PAM-distal end (Szczelkun *et al.*, 2014; Mekler *et al.*, 2017; Ivanov *et al.*, *PNAS*, 2020). In this process, Cas9 does not have a mechanism to “sense” complementarity between the guide RNA ahead of the forming R-loop (in other words, the Cas9-guide RNA complex does not “know” a priori whether a substrate DNA is on-target or off-target until mismatches in the target DNA strand are encountered). The positioning of the PAM-distal end of the R-loop, as seen in the 6-nt and 8-nt match structures thus likely reflects on-pathway intermediates.

The question of R-loop formation in the Cas9 system has been extensively studied using a range of biophysical studies including bulk-solution and single-molecule FRET, magnetic tweezers, AFM and others. Many FRET studies have used the same guide RNA (based on the

lambda1 sequence) as used in our study, and thus their conclusions can be directly extrapolated to our structural insights (Sternberg *et al.*, *Nature*, 2015; Sing *et al.*, *Nat Comm*, 2016; Dagdas *et al.*, *Sci Adv*, 2017; Chen *et al.*, *Nature*, 2018; Ivanov *et al.*, *PNAS*, 2020). These studies collectively point to the existence of a conformational equilibrium checkpoint that senses complementarity of the guide RNA to the target DNA in the PAM-distal part of the R-loop, when the R-loop reaches ~16-17 bp of complementarity. Below 18 bp of complementarity, the HNH, REC2 and REC3 domains predominantly adopt inactive (i.e. checkpoint) conformations, whereas the active conformations predominate beyond 18 bp of complementarity. Notably all these biophysical studies have also relied on the introduction of mismatches in the guide RNA-target DNA heteroduplex to arrest the expanding R-loop in a given state.

In agreement with the conformational checkpoint equilibrium, a previous structural study of the fully complementary Cas9 ternary complex, revealed a mixed population comprising pre-cleavage (i.e. checkpoint), post-cleavage, and product states in the same cryo-EM sample (Zhu *et al.*, NSMB, 2019). Our structural analysis of intermediate R-loop states is fully consistent with these structural insights insofar that the most populated states observed for in the 12-, 14-, 16-nt match complexes are highly similar to the pre-cleavage conformation of the fully paired complex. In the revised manuscript, we now present additional structures of an 18-nt match complex in the pre-cleavage conformation, in which the HNH domain is in an inactive state, as well as an active conformational state (obtained in the presence of Mg²⁺ ions), in which the REC3 and REC2 domains reposition, enabling the HNH domain to engage the target DNA strand and the displaced NTS to bind in the RuvC domain active site.

Additionally, we would like to note that our structural observations are consistent with very recent structural data from the Doudna (Cofsky *et al.*, *bioRxiv*, 2021; <https://www.biorxiv.org/content/10.1101/2021.09.06.459219v1>) and Taylor labs (Bravo *et al.*, *Nature*, 2021), and complement their respective structural analyses of early and late steps in the R-loop formation mechanism. as well as with molecular dynamics simulations of the HNH conformational activation (Palermo *et al.*, *Q Rev Biophys*, 2018; Casalino *et al.*, *ACS Catal*, 2020).

Finally, investigating R-loop formation in real time using single-molecule studies without the need for introducing mismatches, to determine whether these checkpoint states are truly on-pathway, would require considerable increase in the temporal resolution of the measurements (<0.1 s) to account for the fast hybridization kinetics (Sing *et al.*, *Nat Comm*, 2016). Likewise, the current methods for sample preparation in cryoEM microscopy simply cannot reach the required temporal resolution. We thus agree that this is an important question that would warrant further investigation, but these experiments are beyond our means and in our view fall outside of the scope of the current manuscript.

The authors also suggest that the structures establish a set of amino acid residues as good targets for mutations that might increase Cas9 specificity against mismatches. These residues contact the PAM-distal DNA duplex backbone during the transition from the 5-bp R-loop to the 8-bp R-loop, and the suggestion is that mutations in them would further destabilize an intermediate state to promote dissociation of off-target sequences and improve specificity. First, it is unclear what intermediate is being referred to, because if the contacts are not formed until the R-loop is 8 bp, mutating these amino acids would not be expected to destabilize the intermediate with the shorter R-loop. Second, regardless of which intermediate is being considered, the logic of these mutations improving specificity is not clear, because the contacts are apparently formed in the context of a matched duplex, so the simplest expectation is that mutations would

destabilize the matched and mismatched complexes equally, most simply with no impact on specificity. It is possible that there is a more nuanced suggestion to be made regarding the specific kinetic steps in R-loop formation that are specificity determining, but the point is not at all clear in its current form.

We thank the Referee for these comments. Previous kinetic studies and theoretical analyses have shown that Cas9 discriminates on- and off-target substrates based on kinetic partitioning between conformational checkpoint activation (followed by DNA cleavage) and substrate DNA dissociation (Gong *et al.*, *Cell Rep*, 2018, Bisaria *et al.*, *Cell Syst*, 2017). Indeed, many high-fidelity Cas9 variants do not discriminate off-targets by weakening their binding per se (by an increased intrinsic rate of dissociation) but rather by slowing the rate of conformational activation and resulting cleavage, thereby giving the off-target more time to dissociate (Liu *et al.*, *Nat Comm*, 2020). Although mutations that destabilize the R-loop are expected to destabilize both the on- and off-target states equally, they would nevertheless be expected to lead to increased off-target discrimination because of the additional destabilizing effect of the off-target mismatches on the kinetics and thermodynamics of the guide RNA-(off-)-target DNA hybridization process, which exacerbates the destabilizing effect of the protein mutations. We have recently shown that similar results can be obtained using CRISPR hybrid RNA-DNA (chRDNA) guides that contain deoxynucleotides in the spacer segment (Donohoue, Pacesa *et al.*, *Mol Cell*, 2021). These slow down the forward rate of conformational activation while simultaneously increasing the rate of the reverse process of substrate DNA dissociation for both on- and off-target substrates. However, the effect is exacerbated due to the destabilizing effect of the off-target mismatches on the R-loops structure, resulting in enhanced kinetic partitioning and off-target discrimination.

To validate our structural observations, we mutated amino acid residues involved in contacting the R-loop in the 6-, 8- and 10-nt intermediate states, and assessed the nuclease activities of the mutant Cas9 proteins with on- and off-target DNA substrates (data shown in Fig. 1e and Extended Data Fig. 6).

We observe that mutation of the Y450 residue splitting the seed segment of the guide RNA into two parts resulted in significantly decreased levels of off-target cleavage (shown in Fig. 1e). This mutation most likely disrupts the pseudohelical preordering of the seed and perturbs the energetics and kinetics of R-loop formation, and this effect is exacerbated for off-target substrates, leading to increased discrimination.

In addition, the mutation of REC2 residues involved in PAM-distal duplex stabilization reduces the level of off-target discrimination, as described previously (Sung *et al.*, *JACS*, 2018), while REC3 mutations had only marginal effect on Cas9 activity (Extended Data Fig. 6). These results imply that the REC2 domain, and its interdomain interactions contributes to Cas9 specificity.

Finally, concerning residues involved in REC2/REC3 interdomain interactions upon formation of the 10-nt match complex (i.e. when seed pairing is complete), we show that mutations of the REC2 DDD helix residues increase specificity, while mutations in the REC3 RRR helix do not have a strong effect. These results suggest that the 10-nt complex may be destabilized by REC2 DDD mutant, which leads to improved off-target discrimination due to increased kinetic partitioning.

Referee #2:

The authors determined a series of structures of SpyCas9 in complex with a sgRNA and partially matched DNA substrates that contain 6, 8, 10, 12, 14 and 16 complementary nucleotides. The authors used incomplete complementary targets to prevent further extension of the guide-target heteroduplexes to capture different states of the Cas9 along target DNA binding. These structures show, like a movie, the conformational changes of Cas9 and guide-target heteroduplexes during target DNA binding. In particular, they revealed detailed rearrangements of the Rec2, Rec3 and HNH structural domains along their DNA binding pathways. They found that incomplete target pairing does not induce the conformational displacement required for nuclease structural domain activation, explaining why the non-target complex remains trapped in the inactive state. However, although this paper resolves the several structures of different intermediates and reveals structural rearrangement during DNA binding, it does not provide important insights into the molecular mechanisms of on/off-target discrimination.

We thank the Referee for the critical feedback, which has made us make additional revisions to the manuscript to strengthen our conclusions and present the data more clearly.

We respectfully disagree with the statement that our structural studies do not provide important clues about on/off-target discrimination. Our studies are highly complementary, and largely consistent with a large body of biochemical, biophysical and computational studies that have identified the existence of a conformational transitions and checkpoints that sense R-loop integrity and function as major determinants of off-target discrimination. Our study provides critical, high-resolution insights into the structural basis of these conformational transitions and checkpoints, which have remained elusive until now.

To address this issue further and complement our structural studies, we have performed kinetic analysis of structure-guided mutants on fully-matched and off-target substrates, as detailed in our response to Referee #1 above, as well as below.

The following are my major concerns:

1. In the abstract, the authors stated that the incomplete target strand pairing fails to induce the conformational displacements necessary for nuclease domain activation. Do all imperfectly complementary target strands fail to activate Cas9, and if not, what kind of mismatches fail to activate the HNH domain, and what is the effect of the number and position of mismatched nucleotides?

The question of conformational activation has been extensively studied using biophysical studies, including single-molecule FRET analysis (including Sternberg et al., *Nature*, 2015; Sing et al., *Nat Comm*, 2016; Dagdas et al., *Sci Adv*, 2017; Chen et al., *Nature*, 2018; Ivanov et al., *PNAS*, 2020). These studies collectively point to the existence of a conformational equilibrium checkpoint that senses complementarity of the guide RNA to the target DNA in the PAM-distal part of the R-loop, when the R-loop reaches ~16-17-bp of complementarity. With substrate DNAs with <18-bp of complementarity, the HNH, REC2 and REC3 domains predominantly adopt inactive (i.e. checkpoint) conformations, whereas the active conformations predominate beyond 18-bp of complementarity.

We would also like to point attention to our recent preprint (Pacesa *et al.*, *bioRxiv*, 2021) where we report crystal structures of the Cas9 complex bound to 15 genomic off-targets with varying types and positionings of mismatches. We observe various levels of flexibility of the REC2/3 domains, suggesting our crystal structures can capture relevant conformational changes induced by the presence of mismatches. Notably, the conformation of the HNH domain in all the crystal structures containing off-targets is analogous to the 16-nt match and 18-nt match (checkpoint) structures, suggesting that bona fide off-targets progress through similar steps of conformational activation outlined in our manuscript.

Lastly, a complementary study (Bravo *et al.*, *Nature*, 2021) detailing the cryo-EM structures of Cas9 bound to substrates with internal mismatches has revealed that off-targets fail to result in a “kinked” conformation of the heteroduplex necessary for conformational cleavage activation, such as observed in our 18-nt match (catalytic) complex. This could be due to the mismatches distorting the conformation of the heteroduplex resulting in suboptimal relative positioning of the cleavage domains, as suggested by molecular dynamics studies (Zeng *et al.*, *Phys Chem Chem Phys*, 2018; Ricci *et al.*, *ACS Cent Sci*, 2019; Mitchell *et al.*, *Front Mol Biosci*, 2020).

2. Mismatches between the guide and target in the seed region abolish Cas9 DNA cleavage, while mismatches in other region have less effect, resulting in off-target cleavage. The mismatches may affect R-loop formation and decrease or abolish Cas9 DNA cleavage activity. Do single or double nucleotide mismatches in the seed region abolish the activity by preventing R-loop formation? How many nucleotides mismatches block the guide-target heteroduplex propagation? How does the mismatch in different position have different effect?

This question has been extensively addressed by previous high-throughput kinetic studies of on- and off-target substrate cleavage, for example those from the Finkelstein (Jones *et al.*, *Nat Biotech*, 2021) and Greenleaf (Boyle *et al.*, *PNAS*, 2017; Boyle *et al.*, *Sci Adv*, 2021) labs, confirming that the majority of off-target sites are stably bound but not cleaved. These studies have collectively highlighted the importance of seed-sequence matches for specificity and the relative tolerance of mismatches in the PAM-distal part of the guide-TS heteroduplex, which are in good agreement with *in vivo* editing data. Overall, these studies show that single-nucleotide mismatches slow the rate of DNA cleavage by ~100-fold, while two seed mismatches are largely deleterious and prevent DNA cleavage. We also present quantitative data on the kinetics of off-target DNA cleavage in our recent structural analysis of Cas9 off-target activity (Pacesa *et al.*, *bioRxiv*, 2021).

3. The TS and NTS remain hybridized at the PAM-distal end of the DNA. How many base pairs between are observed in 6-nt, 8-nt and 10-nt match structures? The authors found the relocation of the PAM-distal duplex in distinct structures. The superposition of the PAM-distal duplex is needed to show its conformational changes.

The atomic models of the 6-, 8- and 10-nt match cryo-EM structures have 14, 11 and 9 base pairs modeled in the PAM-distal TS-NTS duplexes. We now show a superposition of the duplexes in Fig. 2a.

4. Page 5: The authors found that the 6-nt complex only forms a 5-bp guide-target duplex while the last base pair stacks with Y450, and in the structure of the 8-nt complex R-loop heteroduplex past Tyr450. The authors compared the structures of the 6-nt and 8-nt complexes and concluded that “These observations suggest that the seed sequence of the Cas9 guide RNA

is bipartite and that its hybridization with target DNA proceeds in two step..." How do the heteroduplex lengths and mild conformational changes indicate that the seed sequence of the guide RNA is bipartite? The data here are somewhat over-interpreted.

We thank the Referee for the critical feedback but we respectfully disagree with the statement that the structural data is somewhat overinterpreted with regards to the bipartite seed. The 6-nt match complex structure clearly shows that the seed sequence is interrupted by the insertion of the Y450 side chain into the seed base stack, and the seed is thus presented in two pre-ordered helical segments spanning nucleotides 1-5 and 6-10, respectively. These observations imply that the hybridization is a two-step process, since guide-target hybridization is directional, starting from the PAM-proximal end, as supported by biophysical studies. Even though the first five bases of the seed can hybridize to the target without conformational changes, the formation of a continuous guide-target heteroduplex beyond the fifth position requires Y450 to unstack, driven by a conformational rearrangement of the R-loop and REC2 and REC3 domains, which likely results in a kinetic bottleneck and makes the hybridization occur in two steps. This is supported by FRET studies pointing to the existence of a detectable, short-lived intermediate state during target hybridization (Ivanov *et al.*, *PNAS*, 2020; Sung *et al.*, *JACS*, 2018).

The structural observation of a bipartite seed sequence might also explain the discrepancy between the apparent extent of the seed sequence observed in different studies. While CHIP-Seq data from cells suggests that the seed spans 5-6 PAM-proximal nucleotides (Kuscu *et al.*, *Nat Biotech*, 2014; Wu *et al.*, *Quant Biol*, 2014; O'Geen *et al.*, *Nucl Acid Res*, 2015), biochemical experiments indicated that the seed spans 8-12 PAM-proximal nucleotide (Jones *et al.*, *Nat Biotech*, 2020; Boyle *et al.*, *Sci Adv*, 2021).

To test the role of Y450 in seed-dependent target binding, we assessed the catalytic activity of the Y405A Cas9 mutant against on- and off-target substrates (data now presented as Fig. 1e). This mutant displays increased specificity, as reflected in a significant reduction of cleavage activity with off-target substrates, while on target cleavage remains unperturbed. These results suggest that the absence of the stacking interaction introduces a kinetic barrier in the guide RNA-TS DNA hybridization process. In addition, removal of the stacking interaction between Y450 and adjacent bases could destabilize the preordering of the seed region, leading to perturbed R-loop formation. Notably, this mutant displayed considerably enhanced specificity for an off-target substrate containing a strongly distorting mismatch (A-G) mismatch with the seed sequence at the 5th position, consistent with the notion that this residue is critical for the transition between the two seed binding steps during target strand hybridization. Thus, this mutation could serve as a starting point for the development of new high-fidelity Cas9 variants, particularly to address off-target sites with mismatches in the seed region.

Finally, bipartite seed sequences appear to be a feature of other RNA-guided targeting effectors, including the Cascade complex from type I CRISPR systems (see our response to Referee #3 below), as well as Argonaute proteins, as supported by recent data on human Ago2 (Klum *et al.*, *EMBO J*, 2018, <https://doi.org/10.15252/embj.201796474>).

5. In the 6-nt and 8-nt complementary complexes, both Rec2 and Rec3 form interactions with the PAM-distal duplex. For the fully complementary target DNA, do Rec2 and Rec3 also form similar interactions with the PAM-distal region during R-loop extension? That is, is there any intermediate state with the similar interaction along the target DNA binding? Are these hydrogen bonds observed in the 6-nt and 8-nt complementary complexes unique to these off-

target DNAs? On p. 5, "Mutation of these residues, which would further destabilize this intermediate state and thus promote off-target dissociation, presents an opportunity to generate novel high-fidelity SpCas9 variants", the mutation data support this idea are required here.

The REC2 and REC3 domains maintain the same interactions with the PAM-distal duplex in the 10-, 12-, 14-, 16- and 18-nt (checkpoint) match structures (shown in Fig. 3a), and are overall highly similar to the pre-cleavage conformation of the full-match complex (PDB 6O9Z, Zhu *et al.*, *NSMB*, 2019). The 6- and 8-nt match complexes feature unique interactions of the PAM-distal duplex with REC2 and REC3 lobes, some of which are also observed in the recently reported structures of the DNA interrogation complex (Cofsky *et al.*, *bioRxiv*, 2021; <https://www.biorxiv.org/content/10.1101/2021.09.06.459219v1>), particularly with the “helix rolling basic patch” on REC2. These interactions are expected to occur irrespective of whether the DNA is an on- or off-target substrate as they occur in the fully duplexed part of the dsDNA substrate downstream of the R-loop and involve the deoxyribose-phosphate backbones of the duplex strands only.

To support our structural observations, we have tested the activity of structure-based Cas9 mutants in nuclease activity assays. REC2 mutation have perturbed the off-target discrimination of Cas9, resulting in increased off-target cleavage rates, in agreement with previous data implicating the REC2 domain in off-target discrimination (Sung *et al.*, *JACS*, 2018), while REC3 mutations had only marginal effect on Cas9 activity (Extended Data Fig. 6). These residues might only partially contribute to the stability of the R-loop, as evident by the fact that the deletion of the entire REC2 domain (Extended Data Fig. 6) only had a mild effect on Cas9 activity, comparable to that of our mutants. This hypothesis is further supported by the existence of Type II-C Cas9 orthologues that completely lack the REC2 domain.

6. In Fig. 1e (right panel) and Fig. 2a, the direction of the arrows indicating the repositioning of Rec2 is reversed, but these two figures show that the motion of Rec2 is in the same direction. Structural overlay of the REC2 and REC3 domains in the 6-, 8- and 10-nt match complexes may clarify the structural rearrangement of these domains.

We have provided structural superpositions of the REC2/3 domains in the 6-, 8- and 10-nt match complexes in Fig. 2a.

7. Based on the structures in this manuscript, are the authors able to predicate the conformational changes in the Rec lobe of Cas9 lacking the REC2 domain during target binding?

We have made structural superpositions of the 6- and 8-nt SpCas9 complexes with the DNA-bound structures of AceCas9, CdCas9, CjCas9, NmeCas9 (seed-bound state) and SaCas9, all of which lack REC2 domains. We show the superpositions below for the Reviewer’s reference. The superpositions reveal that the 6- and 8-nt R-loop conformations would be compatible with those structures, without resulting in major steric clashes with their REC lobes. The lack of REC2/3 stabilization of the PAM-distal end could also explain the multiple turnover nature of the Type II-C enzymes. However, these interpretations are rather speculative at this point and would need to be supported with actual structural and biophysical information. For this reason, we have refrained from discussion of this point in the manuscript.

Minor:

1. Page 7: Ext. Data Fig. 9b is Ext. Data Fig. 8b.

We thank the Referee for spotting the error. Now Corrected.

2. In Ext. Data Table 3, what do the nucleotides in red indicate?

The color was meant to indicate the spacer sequence. However, the comment no longer applies as we have revised ED Table 3 to provide additional information.

Referee #3:

Pacesa and Jinek have used a cryo-EM approach to study relevant details of R-loop formation and the coupling to conformational activation of Cas9. By a systematic, step-wise increase of the match between guide and target, intermediate stages of the Cas9 domains were obtained at an unprecedented resolution. This is a very well executed, and very well described, story that, as the authors suggest, will provide a solid basis for future engineering of Cas9, to further enhance its specificity and possibly its activity.

We thank the Referee for their positive evaluation of our work and the constructive feedback.

Comments

1) The authors state that off-target formation generally results in enhanced dissociation (in case of PAM-proximal mismatches) or in a stably-bound, cleavage-incompatible state (PAM-distal mismatches). If that would be the complete story, then off-target interactions would not be a major problem for a range of genome editing applications. It would be appropriate if the authors state that early studies reported rather high cases of off-target indel formation, a result of cleavage and NHEJ-based repair. Moreover, the latter activities would make Cas9 a more robust defense system, that could still target potential escape viruses. Please elaborate a bit on this.

We agree with the Reviewer that the questions of off-target activity is an important one in the context of the genome editing applications of Cas9, and the occurrence of off-target indel formation is very well documented in vivo. As off-target substrates contain mismatches with the guide RNA, the energetics and kinetics of guide-(off-target) hybridization become perturbed, as compared to the fully matched substrate. This typically results in weaker binding (due to faster DNA dissociation) and/or slower cleavage, as documented extensively by systematic, high-throughput studies of model off-target studies (Jones *et al.*, *Nat Biotech*, 2021) and Greenleaf (Boyle *et al.*, *PNAS*, 2017; Boyle *et al.*, *Sci Adv*, 2021), as well as by our recent structural and biochemical analysis of bona fide off-target substrates (Pacesa *et al.* *bioRxiv*, 2021; <https://www.biorxiv.org/content/10.1101/2021.11.18.469088v1>). Although our statement may be overgeneralized, biophysical and biochemical studies clearly show the effect of PAM-proximal mismatches have on off-target dissociation (Singh *et al.*, 2016; Ivanov *et al.*, *PNAS*, 2020; Zhang *et al.*, 2020b; Boyle *et al.*, 2021), while PAM-distal mismatches are compatible with a stably-bound, catalytically inactive complex (Sternberg *et al.*, *Nature*, 2015; Chen *et al.*, *Nature*, 2017; Dagdas *et al.*, *Sci Adv*, 2017; Boyle *et al.*, *Sci Adv*, 2021). Nevertheless, many off-target substrates pass the Cas9 conformational checkpoint, resulting in detectable levels of off-target cleavage in vitro and editing in vivo.

The issue here is also partly a question of semantics and the definition of what constitutes an off-target substrate. Most biochemical or biophysical studies consider off-targets to be any incompletely matched DNA sequences and examine the thermodynamics and kinetics of their binding and/or cleavage. In contrast, genome editing studies generally consider off-targets to be sites that undergo detectable editing (>0.1%) in vivo, i.e. only those that pass the conformational checkpoint. We have revised the introduction section of the manuscript to raise these points and to cite the relevant studies.

2) The observation of a bi-partite seed, with a 2-step base pairing process, reminded me of the situation in Cascade, where the seed is split in two segments of 5 and 2-5 nucleotides. Although

I am aware of the major differences of the two systems, I would be curious if the authors consider this an analogous way of step-wise 'quality control'?

We thank the Referee for raising this point. The bipartite nature of the seed sequence in the Cascade complex is well documented in biochemical (Semenova *et al.*, *PNAS*, 2011) and structural studies (Zhao *et al.*, *Nature*, 2014; Jackson *et al.*, *Science*, 2014; Mulepati *et al.*, *Science*, 2014; Xiao *et al.*, *Cell*, 2017). Interestingly, a recent study from the MacRae lab showed that Argonaute proteins, specifically human Ago2, might also rely on a bipartite seed in the guide RNA for target recognition (Klum *et al.*, *EMBO J*, 2018). We have made references to these analogies in the revised version of the manuscript. It is possible that these convergent examples point to an overarching quality control mechanism of RNA-guided effectors, but this is rather speculative at this point.

3) I remember in previous studies that Cas9-based kinking of dsDNA has been reported, at least by the Doudna group. Has this been observed in the structures presented here?

We do observe kinking of the DNA substrate in the proximity of the PAM, due to the separation of the target and non-target DNA strand. The resulting target DNA-guide RNA heteroduplex adopts an A-form conformation, consistent with previously published structures. This is also consistent with the observations of DNA bending in the early binding intermediates recently reported by the Doudna group (Cofsky *et al.*, *bioRxiv*, 2021; see response to Referee #1 above). We and the Taylor lab (Bravo *et al.*, *Nature*, 2021) now additionally show that the conformational cleavage activation of Cas9 requires the heteroduplex to undergo further kinking at the PAM-distal end (Fig. 4).

4) It is appreciated that the authors, apart from just stating that the provided new insights could provide a basis for optimizing the functionality of Cas9, included concrete suggestions. That comes with the risk that greedy reviewers will ask for more. So here we go: it would be great if the authors would actually test the phenotype of a mutant in which (some of) the positively-charged residues of REC2 and REC3 are substituted.

To validate our structural observations and provide initial insights towards optimizing the functionality of Cas9, we have performed nuclease activity assays to test the on- and off-target activity of structure-based mutations of interacting residues identified in the 6-nt, 8-nt and 10-nt match complexes. These results are shown as Fig. 1e and ED Fig. 6.

Firstly, we show that mutation of the conserved Tyr450 results in increased specificity due to reduced rates of off-target cleavage (Fig. 1e). Absence of the tyrosine stacking in the seed most likely destabilizes seed preordering and therefore perturbs R-loop formation, and this effect is exacerbated for mismatch-containing off-target substrates. This mutant presents an opportunity for the development of novel high-fidelity SpCas9 variants. As most off-target sequences are only bound but not cleaved, these variants could prove useful for applications that rely primarily on the fidelity of Cas9 target binding rather than cleavage, for example base editors or CRISPRi/a.

Concerning Cas9 residues contacting DNA in the 6-nt and 8-nt match complex, we observe that REC2 mutations in fact overall reduce the fidelity of Cas9 (Extended Data Fig. 6), as observed in other studies (Sung *et al.*, *JACS*, 2018). On the other hand, we did not observe a significant effect of the REC3 mutations on Cas9 activity, as observed with previously published high-fidelity mutants. This suggests that the PAM-distal contacts may not be

essential for Cas9 target binding but do implicate the REC2 in the specificity of Type II-A Cas9 proteins such as SpCas9. Interestingly, the REC2 mutants exhibit similar activity as the construct completely lacking the REC2 domain.

As for residues involved in forming REC2/REC3 interdomain interactions upon formation of the 10-nt match complex (I.e. when seed pairing is complete), we show that mutations of the REC2 DDD helix residues increase specificity, while mutations in the REC3 RRR helix do not have a strong effect. Again, the REC2 DDD helix mutant could be used as a starting point for the development of new high-fidelity SpCas9 variants.

Reviewer Reports on the First Revision:

Referees' comments:

Referee #1:

The revised manuscript is strengthened by the added DNA cleavage measurements, which test some predictions from the partial R-loop structures and relate the features seen in the structures to the dynamic process of R-loop formation and the specificity against off-target sequences. However, there are several significant issues with the presentation of these experiments, and perhaps with the experiments themselves.

1. The on-target DNA cleavage rate constants shown in Fig. 1e are surprisingly low. It looks by eye that the cleavage rate is approximately 1 min^{-1} , both for the WT SpCas9 and the Y450A mutant. Previous measurements under what seem to be similar solution conditions revealed rate constants of 1 s^{-1} by the HNH domain and 0.2 s^{-1} by the RuvC domain (Gong et al., Cell Rep., 2018), at least an order of magnitude larger than those reported in the current work. I think it is important to investigate this difference because it might impact the specificity calculations. Also, I might have missed it but I couldn't find which DNA strand was labeled in these experiments. This information should be added or made more prominent if it is already there somewhere. It would also be helpful and important to show data from representative time courses for the on-target and select off-target DNAs.

2. The interpretation of the effects of the Y450A mutation on DNA cleavage specificity does not seem clear to me. The mutation seems to improve specificity for the on-target DNA relative to off-target sequences, but this improvement includes off-target₂, which does not have any PAM-proximal mismatches. If the effect of the Y450A mutation is to destabilize the early intermediate(s) that include formation of only a subset of the PAM-proximal 'seed' base pairs, the mutation might be expected to have no effect on off-target₂, because the intermediates at this early stage would be identical for the on-target and off-target sequences.

3. To test the importance of an interaction observed by cryo-EM between helices in the REC2 and REC3 domains, the authors generated alanine mutations of aspartates in REC2 and arginines in REC3 that are proposed to interact electrostatically to mediate the helix-helix interaction. They found that mutations in REC2 DDD helix reduce off-target cleavage activity but mutations in REC3 RRR helix do not reduce this activity. The simple prediction from the hypothesis of important electrostatic interactions would be that the mutations to either helix would block the contact, and so both sets of mutations would impact off-target cleavage activity similarly. Thus, it would seem that the prediction is not met, perhaps suggesting that this helix-helix interaction does not form during R-loop formation in solution or its formation does not affect DNA cleavage specificity. The authors may want to interpret these results in the manuscript.

Referee #2:

In the revised manuscript, the authors provided additional biochemical data to explore the off-target mechanism and 18-nt matched wt SpCas9 complex structures in the pre-catalytic and catalytic states. In the revision, Pacesa et al. reported a series of cryo-EM structures of spCas9 along the R-loop formation. They found that in the early phase of the R-loop formation, the distal end of the target DNA duplex is positioned in the positively charged cleft formed by the Cas9 REC2 and REC3 domains, consistent with the structure from Taylor (Bravo et al., Nature, 2022). They also reported a bipartite seed in the guide RNA for target recognition by Cas9. Together, this study revealed a DNA-dependent conformational rearrangement of the Cas9 REC-lobe that is necessary for cleavage activation.

I have the following questions:

1. In this study, REC2-deleted Cas9 showed a mild decreased activity (Ext. Data Fig. 6b), while Δ REC2 Cas9 showed a higher on-target cleavage activity than the wild-type Cas9 in the study of Sung et al. (JACS, 2018). Both of the data was obtained through in vitro cleavage assay. However, Δ REC2 Cas9 just kept ~50% editing activity in the study of Nishimasu et al. (Cell, 2014). Can you explain these differences in Δ REC2 Cas9 activity?
2. Do the TS and NTS remain hybridized at the PAM-distal end of the DNA substrate in the 18-nt match complex in Fig. 3b?
3. Page 6, “a subset of mutations of REC2 or REC3 residues resulted in increased off-target cleavage, as did the deletion of the REC2 domain (Extended Data Fig. 6b).”: The authors tested two different off-targets in this assay. These five mutants show similar cleavage activity with wt Cas9 with off-target 4, and R586A/T657A showed lower activity than wt Cas9 with both off-target substrates. Comparing the cleavage dates of two different off-target substrates, I did not find any common pattern in the cleavage activity of these mutants. Would these mutants exhibit different activity if a different non-target DNA was used?
4. In the Ext. Data Fig. 6c, the authors stated that “The off-target cleavage activity of Cas9 in vitro was significantly reduced by alanine substitutions of the interacting residues in the REC2 DDD helix, but not by mutations in the REC3 RRR helix”. Although one REC2 DDD helix mutant showed reduced activity with off-target 2, it just exhibited a modest lower activity than the REC3 mutant and wt using off-target 4. It is hard to get a general rule for the off-target cleavage based on the data in this figure.
5. Bravo et al. (Nature, 2022) reported a fully active SpCas9 complex with 18- to 20-nt mismatched DNA (PDB: 7S4X). Could you compare your on-target state (18-nt matched catalytic structure) with off-target state (PDB: 7S4X) and give any new insight into the 18- to 20-nt off-target mechanism?
6. Page 8, “enabling the HNH domain to undergo a $\sim 140^\circ$ rotation to engage the TS scissile phosphate with its active site and catalyse its hydrolysis via a one-metal-ion mechanism (Fig. 4d), in agreement with prior structural data²⁸.”: Although three structures of SpCas9 were reported in ref. 28, the HNH structural domains in these structures, especially the catalytic center, were of low

resolution and the metal ions could not be identified. Thus, this reference cannot provide the evidence that HNH cleaves DNA via a one-metal-ion mechanism. However, the electron density of HNH is significantly better in the structures of St1Cas9 (Zhang et al., Nat. Catal., 2020) and NmeCas9 (Sun et al., Mol. Cell, 2019), and these two studies may provide strong evidence.

Referee #3:

All questions raised have been addressed appropriately. Congrats with this beautiful study!

Author Rebuttals to First Revision:

Manuscript # 2021-09-14915A

Response to Referees' comments

Referee #1:

The revised manuscript is strengthened by the added DNA cleavage measurements, which test some predictions from the partial R-loop structures and relate the features seen in the structures to the dynamic process of R-loop formation and the specificity against off-target sequences. However, there are several significant issues with the presentation of these experiments, and perhaps with the experiments themselves.

1. The on-target DNA cleavage rate constants shown in Fig. 1e are surprisingly low. It looks by eye that the cleavage rate is approximately 1 min⁻¹, both for the WT SpCas9 and the Y450A mutant. Previous measurements under what seem to be similar solution conditions revealed rate constants of 1 s⁻¹ by the HNH domain and 0.2 s⁻¹ by the RuvC domain (Gong *et al.*, *Cell Rep.*, 2018), at least an order of magnitude larger than those reported in the current work. I think it is important to investigate this difference because it might impact the specificity calculations. Also, I might have missed it but I couldn't find which DNA strand was labeled in these experiments. This information should be added or made more prominent if it is already there somewhere. It would also be helpful and important to show data from representative time courses for the on-target and select off-target DNAs.

We thank the Referee for bringing up the discrepancy between the cleavage rates measured in our activity assays and those reported previously in the Gong *et al.*, *Cell Rep.*, 2018, study.

We were unable to ascertain from the Gong *et al.* publication whether the disparity in cleavage rates could be due to difference in the experimental conditions employed in the two studies, as the exact details of the nuclease activity experiments (time, buffer composition) are not fully reported in the methods section. Furthermore, the discrepancy could come from the way the data is fitted. Gong *et al.* utilised a stopped-flow approach to be able to measure fast kinetics and fit only the very early data points to extract kinetic values (probably only the first 60 s, based on the figures). We fit the cleavage data across multiple hours, to keep consistent fits with the off-target substrates that support much slower cleavage rates.

However, experimental setup differences aside, we believe that the principal reason for the observed differences in the cleavage rates is that our guide RNA sequence (TRAC) differs from the one used in Gong *et al.* ($\lambda 1$). In our recent study of off-target interactions (Pacesa *et al.*, *bioRxiv*, 2021; <https://www.biorxiv.org/content/10.1101/2021.11.18.469088v1>), where we kinetically profiled four different guide RNAs against on- and off-target substrates, we observed up to ~10-fold difference in on-target cleavage rates between the different guides. This was also observed in another recent study (Boyle *et al.*, *Sci. Advances*, 2021), where different guide RNAs exhibited up to 30-fold differences in on-target cleavage rates. Such differences are not surprising and in fact to be expected, given that on-target substrate cleavage rate is primarily determined by the rates of substrate DNA unwinding and R-loop formation (the main conclusion of Gong *et al.*), which in turn depend on the precise energetics of the guide-target hybridization and are therefore sequence-specific.

Concerning the strand labelling strategy in our experiments - only the target DNA strand was labelled on the PAM-proximal end - we have now indicated this in the figures and the Methods section. We have also added graphs representing the time courses of mutant cleavage.

2. The interpretation of the effects of the Y450A mutation on DNA cleavage specificity does not seem clear to me. The mutation seems to improve specificity for the on-target DNA relative to off-target sequences, but this improvement includes off-target2, which does not have any PAM-proximal mismatches. If the effect of the Y450A mutation is to destabilize the early intermediate(s) that include formation of only a subset of the PAM-proximal 'seed' base pairs, the mutation might be expected to have no effect on off-target2, because the intermediates at this early stage would be identical for the on-target and off-target sequences.

The specificity of Cas9 is determined by the kinetic partitioning between the forward reaction, (i.e. R-loop formation, followed by conformational activation at the checkpoint step, and finally cleavage) and the reverse reaction (i.e. R-loop collapse and DNA dissociation), as revealed by previous kinetics studies and theoretical analyses (Gong *et al.*, *Cell Rep*, 2018, Bisaria *et al.*, *Cell Syst*, 2017). For on-target substrates, the kinetic partitioning is heavily biased in the forward direction, as the rate of Cas9 cleavage is primarily determined by DNA strand unwinding and R-loop formation.

Previously developed high-fidelity mutants, such as HiFiCas9, SpCas9-HF1, or HypaCas9 (Vakulskas *et al.*, *Nat. Med.* 2018; Kleinstiver *et al.*, *Nature*, 2016; Chen *et al.*, *Nature*, 2017), where the mutated charged residues are in the vicinity of the PAM-distal end, improve the specificity primarily by slowing down the overall cleavage rate of Cas9 (Liu *et al.* 2020, *Nature Comm.*), thus biasing the kinetic partitioning towards off-target dissociation without affecting the dissociation rate *per se*.

Although our structures reveal that Y450 plays a structural role in stabilizing the early binding intermediate states, Y450 already interacts with the seed sequence of the guide RNA in the initial state (i.e. binary Cas9-guide RNA complex), likely facilitating its preordering. The Y450A mutation might thus have a complex effect on the thermodynamics and kinetics of R-loop formation. Although it might be expected that a perturbation of the seed sequence preordering would equally affect the kinetics of R-loop formation for both on-target and off-target substrates, our data indicates that the mutation is not sufficient to perturb the rate limiting step in on-target substrate cleavage. However, the destabilising effect is greatly exacerbated in the case of off-target substrates, which already have a defect in either R-loop formation or checkpoint activation (as evidenced by their lower cleavage rate constants with WT Cas9 in Fig. 1e), resulting in increased off-target discrimination. Although off-targets with no seed sequence mismatches are also discriminated against (~6.2-fold reduction in off-target #2 cleavage, ~5-fold for off-target #5), the effect is more prominent for off-targets containing seed-sequence mismatches (up to ~29-fold for off-target #4), which is entirely consistent with our structural observations. Furthermore, the results are fully consistent with previous data showing that the Y450A mutation enhances off-target discrimination, particularly for seed mismatches (Kleinstiver *et al.*, *Nature*, 2016).

Overall, this suggests that the Y450A mutation likely results in faster DNA dissociation rate, although further work, including single-molecule experiments, would be necessary to pinpoint the specificity enhancement mechanism. However, we believe that these experiments are outside the scope of the present manuscript.

To clarify this point, we have revised our manuscript to state:

“Alanine substitution of Tyr450 resulted in substantial reductions of off-target substrate cleavage, while on-target cleavage remained largely unperturbed (**Fig. 1e**). As observed previously³², the effect was more prominent for off-target substrates containing mismatches with the seed region of the guide RNA when compared to off-targets containing only PAM-distal mismatches. Together, these results suggest that disruption of seed sequence interactions in the binary Cas9-sgRNA complex and early binding intermediates might exacerbate R-loop destabilisation caused by off-target mismatches, resulting in increased rate of off-target substrate dissociation and thus increased specificity. In contrast, a subset of mutations of DNA-interacting REC2 or REC3 residues resulted in increased off-target cleavage, as did the deletion of the REC2 domain (**Extended Data Fig. 6b**), consistent with prior single-molecule studies implicating the REC2 domain in Cas9 specificity³¹.”

3. To test the importance of an interaction observed by cryo-EM between helices in the REC2 and REC3 domains, the authors generated alanine mutations of aspartates in REC2 and arginines in REC3 that are proposed to interact electrostatically to mediate the helix-helix interaction. They found that mutations in REC2 DDD helix reduce off-target cleavage activity but mutations in REC3 RRR helix do not reduce this activity. The simple prediction from the hypothesis of important electrostatic interactions would be that the mutations to either helix would block the contact, and so both sets of mutations would impact off-target cleavage activity similarly. Thus, it would seem that the prediction is not met, perhaps suggesting that this helix-

helix interaction does not form during R-loop formation in solution or its formation does not affect DNA cleavage specificity. The authors may want to interpret these results in the manuscript.

We respectfully disagree with the proposition that the REC2 DDD-REC3 RRR interactions would not form in solution. Cryo-EM structures are obtained from rapidly vitrified samples without the need for crystallization and thus represent the native conformations of the respective macromolecules in solution. Furthermore, the DDD and RRR helices are located in proximity of each other in the majority of published Cas9 cryo-EM and crystal structures (PDB codes: 4UN3, 4OO8, 5F9R, 6K4P, 7S4X), suggesting that this is not an artefact of the sample preparation.

Although both the DDD and RRR mutations are expected to perturb the REC2-REC3 domain interaction during R-loop formation, one has to consider that these sextuple mutations change the net charge of the Cas9 protein in solution (DDD by +6, RRR by -6) and therefore also affect its solvation and non-specific (i.e. charge-driven) interactions with nucleic acids. For these reasons, the overall physicochemical consequences of the DDD or RRR mutations on Cas9 activity might be more complex and should in fact be expected to have non-equivalent effects on the on-/off-target cleavage activities.

Furthermore, although the REC2-REC3 interaction mediated by the RRR and DDD helices facilitates the conformation transition in Cas9 induced by R-loop extension, the helices might play additional roles during upstream and downstream steps in the mechanism. In particular, the RRR helix contacts the backbone of the PAM-distal duplex in the 6-nt and 8-nt match complexes.

To address these points, we have revised the relevant section of the manuscript (p. 7) to state: "Cleavage of off-target substrates in vitro was significantly reduced by alanine substitutions of the interacting residues in the REC2 DDD helix, whereas mutations in the REC3 RRR helix only reduced cleavage of the off-target substrate containing a mismatch in the seed region (**Extended Data Fig. 6c**). These results suggest that the REC2-REC3 interaction contributes to Cas9 restructuring during R-loop extension beyond the seed region, although the DDD and RRR helices might also play additional structural roles during upstream and downstream steps in the DNA binding mechanism, particularly as the RRR helix contacts the backbone of the PAM-distal DNA duplex during early stages of target binding (**Extended Data Fig. 5d,h**)."

Referee #2:

In the revised manuscript, the authors provided additional biochemical data to explore the off-target mechanism and 18-nt matched wt SpCas9 complex structures in the pre-catalytic and catalytic states. In the revision, Pacesa et al. reported a series of cryo-EM structures of spCas9 along the R-loop formation. They found that in the early phase of the R-loop formation, the distal end of the target DNA duplex is positioned in the positively charged cleft formed by the Cas9 REC2 and REC3 domains, consistent with the structure from Taylor (Bravo et al., Nature, 2022). They also reported a bipartite seed in the guide RNA for target recognition by Cas9. Together, this study revealed a DNA-dependent conformational rearrangement of the Cas9 REC-lobe that is necessary for cleavage activation.

I have the following questions:

1. In this study, REC2-deleted Cas9 showed a mild decreased activity (Ext. Data Fig. 6b), while Δ REC2 Cas9 showed a higher on-target cleavage activity than the wild-type Cas9 in the study of Sung et al. (JACS, 2018). Both of the data was obtained through in vitro cleavage assay. However, Δ REC2 Cas9 just kept ~50% editing activity in the study of Nishimasu et al. (Cell, 2014). Can you explain these differences in Δ REC2 Cas9 activity?

We thank the Referee for bringing up this point. We are aware of this inconsistency. In the Δ REC2 construct that we used, we replaced the entire REC2 domain with a flexible GGSGGS linker, same as was done in Sung et al., JACS, 2018. We note that the Δ REC2 Cas9 protein

had lower stability during purification, which could result in lower specific activity in our bulk experiments, and would be consistent with the reduction in editing efficiency observed in the Nishimasu *et al.* study. In contrast, the Sung *et al.* study derived the cleavage kinetics parameters from single-molecule fluorescence experiments, in which the Cas9 protein is in large excess over the DNA substrate.

2. Do the TS and NTS remain hybridized at the PAM-distal end of the DNA substrate in the 18-nt match complex in Fig. 3b?

We observe residual densities at the PAM-distal end both in the 18-nt match checkpoint and catalytic complexes. However, the densities are only visible at low contour levels, indicating structural flexibility of the PAM-distal part of the R-loop, and therefore too diffuse to permit accurate model building to define the extent of the guide-target and the PAM-distal TS-NTS duplexes. In the 18-nt catalytic complex, the density is sufficiently large for us to be able to conclude that the TS and NTS remain hybridized at the PAM-distal end and coaxially stacked with the guide-TS duplex. The guide RNA and TS are paired at the 19th position via non-canonical base-pairing, but we are unsure about base pairing at the 20th position. In the 18-nt match checkpoint complex, we can only confidently build a 18-bp guide-TS duplex. There is additional density beyond it but it is not clear whether this represents two additional non-canonical base pairs of the guide-TS duplex, or a coaxially stacked TS-NTS duplex.

3. Page 6, “a subset of mutations of REC2 or REC3 residues resulted in increased off-target cleavage, as did the deletion of the REC2 domain (Extended Data Fig. 6b).”: The authors tested two different off-targets in this assay. These five mutants show similar cleavage activity with wt Cas9 with off-target 4, and R586A/T657A showed lower activity than wt Cas9 with both off-target substrates. Comparing the cleavage dates of two different off-target substrates, I did not find any common pattern in the cleavage activity of these mutants. Would these mutants exhibit different activity if a different non-target DNA was used?

We would not necessarily expect to see a clear pattern, since each off-target has a unique kinetic and thermodynamic hybridization profile as a result of the underlying sequence and the positions and types of mismatches. This is supported by our recent structural investigations of Cas9 off-target activity (Pacesa *et al.*, *bioRxiv*, 2021; <https://www.biorxiv.org/content/10.1101/2021.11.18.469088v1>), where we observe that the structural mechanisms of mismatch tolerance are highly dependent on the type and position of the off-target mismatches, as well as their sequence context. Some off-targets induced allosteric rearrangements of the REC2 and REC3 domains relative to each other. Different REC2/REC3 domain mutants might thus reduce off-target cleavage to a different degree.

4. In the Ext. Data Fig. 6c, the authors stated that “The off-target cleavage activity of Cas9 *in vitro* was significantly reduced by alanine substitutions of the interacting residues in the REC2 DDD helix, but not by mutations in the REC3 RRR helix”. Although one REC2 DDD helix mutant showed reduced activity with off-target 2, it just exhibited a modest lower activity than the REC3 mutant and wt using off-target 4. It is hard to get a general rule for the off-target cleavage based on the data in this figure.

As stated in our response to a similar comment from Referee #1, the DDD and RRR helix mutations would be expected to have non-identical effects on Cas9 activity due to their diametrically opposite effects on the net charge on the Cas9 protein. Furthermore, the DDD and RRR helices might also play additional structural roles during upstream and downstream steps in the DNA binding mechanism. We have revised the relevant part of the text to provide additional interpretation of the results.

5. Bravo *et al.* (Nature, 2022) reported a fully active SpCas9 complex with 18- to 20-nt mismatched DNA (PDB: 7S4X). Could you compare your on-target state (18-nt matched catalytic structure) with off-target state (PDB: 7S4X) and give any new insight into the 18- to 20-nt off-target mechanism?

We would like to note that our 18-nt match complex structures contains 18 fully-formed gRNA-DNA base pairs, while the 18-20MM structure reported in Bravo *et al.* contains 17 base pairs.

When overlaid, the two structures show highly similar domain conformations, with only slightly different orientations of their respective REC2 domains.

We do observe notable differences at the PAM-distal end of the R-loop. In the Bravo et al. structure, a RuvC domain loop (residues 1013-1030) assumes an ordered conformation as it inserts into the mismatched end of the guide-TS heteroduplex, establishing backbone interactions with the TS. Although not mentioned in the Bravo et al. paper, the ordering of the 1013-1030 loop appears to be facilitated by nucleotides 19-23 of the displaced NTS, which forms a stem loop due to intrastrand base-pairing within the NTS, thus helping to constrain the 1013-1030 loop in its ordered conformation.

In our structure, the heteroduplex contains 18 canonical base pairs as well as a stacked rA-dA base mispair at the 19th position, which stabilizes the sgRNA backbone in the proximity of the REC3 and RuvC domains. In contrast to the 18-20MM structure, residues 1014-1029 of the RuvC loop remain disordered.

One reason for the observed differences is that the RuvC loop contributes to the activation checkpoint by transiently interacting with the 17-bp heteroduplex, and then becoming disordered once the checkpoint is passed and additional base pairs form at the PAM-distal end of the guide RNA-TS DNA heteroduplex.

However, in our view, it is equally plausible that the ordering of the 1013-1030 loop observed in the 18-20MM complex structure only occurred as a result of the specific choice of the off-target sequence, because it is facilitated by the sequence-specific structure adopted by the displaced NTS. With a different set of mismatches at positions 18-20, the loop might have also remained in the disordered state. Bravo et al. show that mutations in the RuvC loop dramatically diminish cleavage of the same 18-20MM off-target sequence used for structure determination, but unfortunately did not show data for any other off-target sequences.

Additional structures and biochemical experiments would thus be needed to resolve the differences observed in our 18-nt match and the Bravo *et al.* 18-20MM structures. However, we believe that these fall outside of the scope of the present manuscript and should be the focus of a follow-up study.

6. Page 8, “enabling the HNH domain to undergo a ~140° rotation to engage the TS scissile phosphate with its active site and catalyse its hydrolysis via a one-metal-ion mechanism (Fig. 4d), in agreement with prior structural data²⁸.”: Although three structures of SpCas9 were reported in ref. 28, the HNH structural domains in these structures, especially the catalytic center, were of low resolution and the metal ions could not be identified. Thus, this reference cannot provide the evidence that HNH cleaves DNA via a one-metal-ion mechanism. However, the electron density of HNH is significantly better in the structures of St1Cas9 (Zhang et al., Nat. Catal., 2020) and NmeCas9 (Sun et al., Mol. Cell, 2019), and these two studies may provide strong evidence.

We thank the Referee for the suggestions. We agree that the listed papers are more appropriate references for the one-metal ion catalytic mechanism of the HNH domain. We have included them in the main text.

Referee #3:

All questions raised have been addressed appropriately. Congrats with this beautiful study!

We thank the Referee for their comments. We are happy that we were able to address them and that the suggested revisions improved the manuscript.

Reviewer Reports on the Second Revision:

Referees' comments:

Referee #1:

The authors have addressed my concerns, and I congratulate them on the work. As a remaining very minor issue, it would be helpful to show the best-fit curves to the DNA cleavage data in Ext. Data Fig. 6, whereas the points are currently connected arbitrarily by lines. Also, it would be helpful to similarly show the time course data for the Y450A mutant here or in an additional Ext. Data figure. These data are currently included as an Excel table but not presented as a graph (unless I missed it somehow).

Referee #2:

I have only concerns about the statistics of the structure. According to Ext. Data Table 1, protein B factors of models 7Z4G, 7Z4H, and 7Z4I are above 200 Å², while the resolution is between 3.12-3.64 Å. The values are too high; please explain it. Models 7Z4L, and 7Z4J have the same problem.

Author Rebuttals to Second Revision:

Referee #1:

The authors have addressed my concerns, and I congratulate them on the work. As a remaining very minor issue, it would be helpful to show the best-fit curves to the DNA cleavage data in Ext. Data Fig. 6, whereas the points are currently connected arbitrarily by lines. Also, it would be helpful to similarly show the time course data for the Y450A mutant here or in an additional Ext. Data figure. These data are currently included as an Excel table but not presented as a graph (unless I missed it somehow).

We thank the Referee for the positive comments.

We have revised Extended Data Fig. 6 to include best-fit curves for the DNA cleavage data. We have also added a panel depicting the time course data for the Y450A mutant (now in Extended Data Fig. 6b).

Referee #2

I have only concerns about the statistics of the structure. According to Ext. Data Table 1, protein B-factors of models 7Z4G, 7Z4H, and 7Z4I are above 200 Å², while the resolution is between 3.12-3.64 Å. The values are too high; please explain it. Models 7Z4L, and 7Z4J have the same problem.

We thank the Referee for their comment. We would like to note that atomic B-factors (i.e. Atomic Displacement Parameters, ADPs) of 200 Å² are not unusual for cryo-EM structures at these resolutions. We agree that had these structures been determined from X-ray diffraction data, the B-factor values would certainly have been considered to be too high and indicative of substantial disorder or poor fit in the electron density maps. However, this is not the case for atomic models that have been built and refined using cryo-EM maps. As X-ray and cryo-EM maps depict fundamentally distinct types of density (electron density vs. Coulombic electrostatic potential, respectively) as a result of different physical phenomena (photon vs. electron scattering), the atomic B-factors of models built and refined using X-ray and cryo-EM data are not directly comparable.

The ADP (B-factor) is a well-defined X-ray scattering parameter that describes the fall-off of scattered amplitude as a function of resolution, due to thermal motion of the scattering atoms or their structural disorder. For cryo-EM models, the ADPs are refined against EM maps using a pseudo-crystallographic, reciprocal-space approach (e.g. in Phenix). In this process, the absolute values of ADPs are also affected by effects arising from detector properties, particle alignment errors as well as the specific software package used for data processing.

In consequence, the resulting ADPs are not physically meaningful and the average B-factor of an atomic model based on cryo-EM map is therefore not a suitable indicator of model quality or structural disorder. However, the relative variation of the ADPs within the model can nevertheless be used as a measure of local structural disorder. In general, the question of atomic displacement parameters in cryo-EM structural models is currently unresolved as the appropriate validation metrics for cryo-EM structures are still under active development.

The topic has been extensively reviewed in the literature, for example in:

Law et al., Evolving data standards for cryo-EM structures, *Structural Dynamics* (2020), 7, 014701; <https://doi.org/10.1063/1.5138589>

Masmaliyeva and Murshudov. *Acta Cryst. D* (2019); 75 505–518. doi: [10.1107/S2059798319004807](https://doi.org/10.1107/S2059798319004807);

Pintilie and Chiu. Validation, analysis and annotation of cryo-EM structures, *Acta Cryst. D* (2021). [D77](https://doi.org/10.1107/S2059798321006069), 1142-1152, doi: [10.1107/S2059798321006069](https://doi.org/10.1107/S2059798321006069).